# Double-shelled hollow rods assembled from nitrogen/sulfur-codoped carbon coated indium oxide nanoparticles as excellent photocatalysts

Liming Sun [1,3], Rong Li[1,3], Wenwen Zhan [1], Yusheng Yuan[1], Xiaojun Wang [1], Xiguang Han[1] & Yanli Zhao [2]

Excellent catalytic activity, high stability and easy recovery are three key elements for fabricating efficient photocatalysts, while developing a simple method to fabricate such photocatalysts with these three features at the same time is highly challenging. In this study, we successfully synthesized double-shelled hollow rods (DHR) assembled by nitrogen (N) and sulfur (S)-codoped carbon coated indium(III) oxide ($In_2O_3$) ultra-small nanoparticles (N,S-C/$In_2O_3$ DHR). N,S-C/$In_2O_3$ DHR exhibits remarkable photocatalytic activity, high stability and easy recovery for oxidative hydroxylation reaction of arylboronic acid substrates. The catalyst recovery and surface area were well balanced through improved light harvesting, contributed by concurrently enhancing the reflection on the outer porous shell and the diffraction in the inside double-shelled hollow structure, and increased separation rate of photogenerated carriers. Photocatalytic mechanism was investigated to identify the main reactive species in the catalytic reactions. The electron separation and transfer pathway via N,S-codoped graphite/$In_2O_3$ interface was revealed by theoretical calculations.

[1] Jiangsu Key Laboratory of Green Synthetic Chemistry for Functional Materials, Department of Chemistry, School of Chemistry and Materials Science, Jiangsu Normal University, Xuzhou 221116, P. R. China. [2] Division of Chemistry and Biological Chemistry, School of Physical and Mathematical Sciences, Nanyang Technological University, 21 Nanyang Link, Singapore 637371, Singapore. [3] These authors contributed equally: Liming Sun, Rong Li. Correspondence and requests for materials should be addressed to X.H. (email: xghan@jsnu.edu.cn) or to Y.Z. (email: zhaoyanli@ntu.edu.sg)

Photoredox catalysis is a highly promising route to solar-to-chemical energy conversion in the form of organic synthesis under relatively mild conditions[1–9]. Among numerous photocatalysts studied in recent decades, indium oxide ($In_2O_3$) with narrow bandgap (~2.8 eV), high optical transparency, excellent stability, and suitable band potentials (including valence band (VB) and conduction band (CB)) to induce redox reactions has attracted much attention as a visible-light-induced photo-catalyst[10–15]. Compared with bulk $In_2O_3$, granular $In_2O_3$ nano-particles have large surface areas, so they contain more active sites and display improved photocatalytic activity[16–18]. However, $In_2O_3$ nanoparticles still possess two undesirable disadvantages, i.e., the fast recombination of photogenerated carriers[19] and the hard recovery in solution, which suppress their practical applications in photocatalytic field. Therefore, a rational design of nanosized granular $In_2O_3$ with high reactivity and easy recovery is highly desirable for extending their applications.

One effective approach to achieve the above-mentioned target is to assemble carbon-coated $In_2O_3$ nanoparticles into the superstructure with regular morphology and large size. On account of excellent mobility of charge carriers, the carbon layer can act as photogenerated electron acceptor to improve the separation efficiency of photogenerated carriers[20–22]. In particular, N,S-codoping sites with induced structure defects in carbon framework could increase the electron delocalization, further enhancing the separation efficiency of electrons and holes[23–25]. Thus, integrating N,S-codoped carbon layer with $In_2O_3$ nano-particles would effectively enhance the photocatalytic activity. Moreover, assembling the resulted N,S-codoped carbon-coated $In_2O_3$ nanoparticles into the superstructure may facilitate the recovery of photocatalysts. However, there are two bottlenecks in this approach for fabricating efficient $In_2O_3$ photocatalysts. Firstly, it is difficult to control the coating of a carbon layer and introduce double heteroatoms into the carbon layer. In general, sufficient and intimate contact interface is the key factor to ensure efficient transfer of photogenerated carriers. However, most of the reported carbon materials as the substrate loaded on metal oxide nanoparticles had insufficient carbon/nanoparticle interface[26]. Uniformly coating a carbon layer on metal oxide nanoparticles is still a challenging task. In addition, conventional approaches for introducing heteroatoms into carbon materials have some una-voidable disadvantages, such as toxic precursors, special and/or sophisticated instruments, harsh conditions, and uneven dis-tribution of doped atoms[27–30]. Realizing the co-doping of two types of heteroatoms is even harder. Therefore, it is highly needed to overcome the challenge toward fabricating ultrafine metal oxide nanoparticles coated by co-doped carbon layers. Secondly, it is difficult to balance the ease of photocatalyst recovery with the surface area. Although assembling N,S-codoped carbon-coated $In_2O_3$ ultrafine nanoparticles into photocatalysts with regular morphology and large size is a straightforward method to solve the problem of difficult recycling[31,32], this solution would result in the decreases of surface areas, reactive sites and photocatalytic activity. Therefore, it is necessary to balance the effects of pho-tocatalyst recovery and surface area by improving other factors affecting photocatalytic activity. To address these issues, we developed a promising approach to fabricate efficient photo-catalysts assembled from N,S-codoped carbon uniformly coated $In_2O_3$ nanoparticles, aiming to achieve excellent catalytic activity and easy recovery at the same time.

Herein, double-shelled hollow rods (DHRs) assembled by N,S-codoped carbon uniformly coated $In_2O_3$ ultrafine nanoparticles (N,S-C/$In_2O_3$ DHR) were synthesized using a porous metal organic framework (MOF), i.e., MIL-68-In, as the template and 1,2-benzisothiazolin-3-one (BIT) as the modulator in one step. Since the carbon atom disperses homogeneously in the MOF

structure at a molecular level[33–42], the carbon layer in N,S-C/$In_2O_3$ DHR is uniformly distributed on the $In_2O_3$ surface to produce intimately contacted core-shell structure and maximize the interaction between the components, facilitating the separa-tion of charge carriers. The obtained N,S-C/$In_2O_3$ DHR could be recycled from the solution through simple centrifugation. Moreover, N,S-C/$In_2O_3$ DHR performs remarkably in catalyzing a series of arylboronic acids under light irradiation. The photo-catalytic time of N,S-C/$In_2O_3$ DHR required to achieve 99% yield was only half of the time needed by recently reported N-doped carbon-coated dodecahedral $In_2O_3$ (N-C/$In_2O_3$ HD)[43]. In addi-tion, the as-prepared N,S-C/$In_2O_3$ DHR exhibits high photo-catalytic activity under green (3 W LED lamp, 525 nm) and yellow (3 W LED lamp, 590 nm) lights, indicating that it broadens the response range of light to wider wavelength. This observation is of great significance for expanding the practical application of $In_2O_3$-based materials. However, N-C/$In_2O_3$ HD did not show any photocatalytic activity under green and yellow lights[43]. These comparison results suggest the importance of N,S-codoping for improving the photocatalytic activity of carbon-coated photo-catalysts. Benefiting from the simultaneously increased reflection on the outer shell and diffraction on the hollow cavity of the N,S-codoped double-shelled hollow structure, and the enhanced separation rate of photogenerated carriers, the photocatalyst recovery and surface area were well balanced. The mechanism for the enhanced separation efficiency of photogenerated carriers by N,S codoping was clarified by theoretical calculations, i.e., the hybridization of N 2p, S 3p and C 2p increases the strength of interstitial states to further improve the transfer and separation efficiency of photogenerated carriers. We also investigated pho-tocatalytic mechanism of the oxidative hydroxylation and con-firmed the key active radical of this system in the photocatalysis. Finally, the microscopic pathway for the separation and migra-tion of carriers in N,S-C/$In_2O_3$ DHR was revealed by theoretical calculations.

## Results

**Structure and composition of the photocatalysts**. The pre-paration of monodispersed double-shelled hollow rods (N,S-C/$In_2O_3$ DHR) assembled by N,S-codoped carbon-coated $In_2O_3$ ultrafine nanoparticles is schematically shown in Fig. 1. The experimental details are presented in the Supporting Information. During the synthetic process, BIT acted a modulator to control the growth of MIL-68-In crystals from indium metal ion and p-phthalic acid ligand, resulting in the formation of N,S-codoped MIL-68-In with the rod morphology. The length-width ratio of N,S-codoped MIL-In-68 rods could be controlled by adding dif-ferent amounts of BIT[44,45]. Upon increasing the amount of BIT, the length-width ratio of N,S-codoped MIL-In-68 rods gradually decreases (Supplementary Fig. 1). A subsequent high-temperature annealing treatment of N,S-codoped MIL-68-In with the largest length-width ratio in vacuum atmosphere leads to the formation of N,S-C/$In_2O_3$ DHR. Typical diffraction peaks in the powder X-ray diffraction (XRD) pattern (Supplementary Fig. 2a) of N,S-codoped MIL-68-In can be well indexed to phase-pure MIL-68-In, which are in good agreement with simulated ones. Scanning electron microscopy (SEM) images of N,S-codoped MIL-68-In at different magnifications are shown in Fig. 2a. The size of N,S-codoped MIL-68-In crystals is 0.4–1.2 μm in width and 5–9 μm in length (Supplementary Fig. 3). Moreover, these N,S-codoped MIL-68-In crystals are highly monodispersed, showing well-defined rod morphology with smooth surface.

Notably, the BIT molecule provides copious N and S elements into MIL-68-In for enabling the creation of N and S codoped materials. To check the existence of N and S elements in

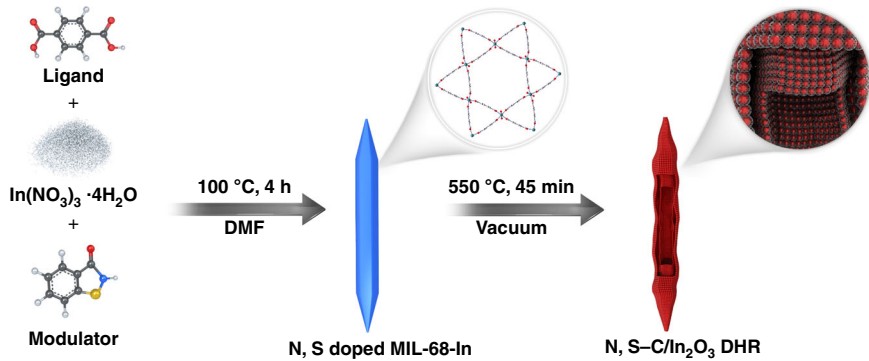

**Fig. 1** Schematic illustration showing the synthetic process of monodispersed N,S-C/In$_2$O$_3$ DHR

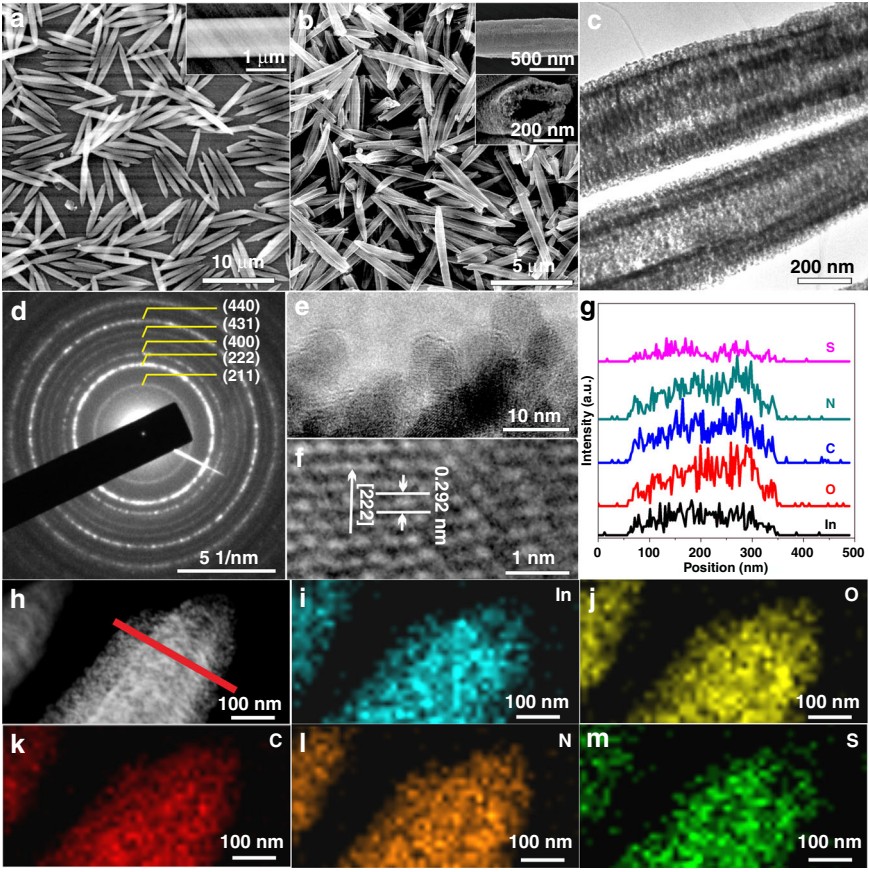

**Fig. 2** Morphology and structural characterizations. **a** SEM image of N,S-codoped MIL-68-In. **b** SEM images, **c** TEM image, **d** Corresponding SAED pattern, **e** Local enlarged TEM image, **f** HRTEM image, **g** Elemental profiles, **h** STEM image, and **i–m** EDX elemental mapping of In, O, C, N, and S for N,S-C/In$_2$O$_3$ DHR. Source data are provided as a Source Data file

MIL-68-In, the energy-dispersive X-ray (EDX) mapping of N,S-codoped MIL-68-In was measured to investigate the element distributions. The corresponding EDX maps of In, O, C, N, and S elements (Supplementary Fig. 4) indicate that all these elements are well dispersed in MIL-68-In. The thermal stability of N,S-codoped MIL-68-In is investigated by thermogravimetric analysis (TGA, Supplementary Fig. 5). According to the TGA curve, initial 6.7% weight lost between 400 and 500 °C can be ascribed to the adsorbed and coordinated solvent molecules, and the steep weight loss (about 55.5%) is due to the carbonization of N,S-codoped MIL-68-In. Hence, annealing N,S-codoped MIL-68-In at 550 °C in a vacuum atmosphere is sufficient to ensure complete conversion from MIL-68-In to In$_2$O$_3$.

Crystalline structure of the annealed product (N,S-C/In$_2$O$_3$ DHR) was explored by powder XRD (Supplementary Fig. 2b). All of the diffraction patterns correspond well with the cubic In$_2$O$_3$ (JCPDS card number 00–006–0416), indicating that the N,S-codoped MIL-68-In precursor is transformed to cubic In$_2$O$_3$ during the annealing treatment. The SEM image (Fig. 2b) indicates that the overall rod morphology of MIL-68-In is well preserved with 0.5–0.8 μm in width and 5–8 μm in length (Supplementary Fig. 6). The size of the annealed product is slightly smaller than that of N,S-codoped MIL-68-In, which is mainly attributed to partial collapse of MOF skeleton and the carbonization of organic ligand and modulator during the transformation process. The magnified observation (insert of

Fig. 2b) reveals the micro-rod morphology with a double-shelled hollow structure, resulted from heterogeneous decomposition of the MIL-68-In MOF during the annealing process. The exterior MOF materials firstly decompose to form an $In_2O_3$ outer shell, and then the inner MOF structure is gradually carbonized under prolonged pyrolysis on account of large temperature gradient from the surface to the inner structure.

More structural details of double-shelled micro-rods are revealed by transmission electron microscopy (TEM). The low magnification TEM image (Fig. 2c) shows obviously brighter contrast, which demonstrates the double-shelled feature and porous nature of the resulting N,S-C/$In_2O_3$ DHR, corresponding well with the SEM observations. The well-defined rings of electron diffraction pattern (Fig. 2d) confirm the polycrystallinity of $In_2O_3$ rods. A high-magnification TEM survey (Fig. 2e) from a typical sample area shows that the porous rod structure of N,S-C/$In_2O_3$ DHR is really constituted by individual nanoparticles with the diameter about 10 nm. The TEM image also clearly exhibits a continuous carbon layer covered on the entire surface of nanoparticles. The surface carbon layer can increase the electrical conductivity of $In_2O_3$ nanoparticles, and thus improves the separation efficiency of photogenerated carriers. In addition, the external carbon layer can block the aggregation and increase the stability of the nanoparticles. Clear lattice fringes were observed from the high-resolution TEM (HRTEM) image (Fig. 2f), where 0.292 nm is assigned to the (222) interplane spacing of cubic $In_2O_3$, further confirming the cubic $In_2O_3$ phase of the nanoparticles. The presence of the In, C, O, S, and N elements in N,S-C/$In_2O_3$ DHR was proven by elemental distribution analysis, including the elemental line profiles (Fig. 2g) and elemental mapping. Scanning transmission electron microscopy (STEM) image indicates that the double-shelled structure of the $In_2O_3$ rods is indeed built by small nanoparticles (Fig. 2h). The elemental mapping results (Fig. 2i–m) demonstrate the uniform distribution of the In, C, O, S, and N within the N,S-C/$In_2O_3$ DHR structure.

During the pyrolysis, the indium ion in the MIL-68-In micro-rods is converted to $In_2O_3$ nanoparticles, and the external N,S-codoped carbon layer is formed by in situ carbonization. Therefore, the final annealed product is N,S-codoped carbon-coated $In_2O_3$ ultra-small nanoparticles with DHR morphology. The porous feature of N,S-C/$In_2O_3$ DHR was investigated by $N_2$ adsorption/desorption isotherms (Supplementary Fig. 7). The Brunauer-Emmett-Teller (BET) measurements give a specific surface area of 37.6 $m^2$/g, and the Barrett-Joyner-Halenda (BJH) pore size distribution is mainly centered at about 10–50 nm (insert of Supplementary Fig. 7). These results confirm that the N,S-C/$In_2O_3$ DHR obtained by the calcination of N,S-codoped MIL-68-In in vacuum atmosphere has DHR morphology with a mesoporous structure.

Chemical states of elements in the N,S-C/$In_2O_3$ DHR were measured by X-ray photoelectron spectroscopy (XPS). The survey spectrum (Fig. 3a) indicates the existence of In, O, N, S, and C elements in N,S-C/$In_2O_3$ DHR. The high-resolution XPS spectrum of In 3d shows two peaks corresponding to In $3d_{3/2}$ and In $3d_{5/2}$ (Fig. 3b). Higher binding energy at 444.5 eV corresponds to In $3d_{5/2}$ of $In_2O_3$, and the peak at 452.1 eV is in agreement with In $3d_{3/2}$ of $In_2O_3$. The high-resolution spectrum of O 1s displays an asymmetric curve (Fig. 3c), which consists of three peaks corresponding to three kinds of oxygen in N,S-C/$In_2O_3$ DHR. The three peaks at 529.8 eV ($O_L$), 530.6 eV ($O_V$) and 531.8 eV ($O_C$) are assigned to lattice oxygen, oxygen-deficient region, and chemisorbed oxygen species (e.g., hydroxyl species), respectively[31,46]. The C 1s high-resolution spectrum (Fig. 3d) can be coherently fitted by five major peaks, assigning to C=O bonds (288.9 eV), C–O bonds (286.8 eV), C–N bonds (285.8 eV), C=C

bonds (284.9 eV) and C–S bonds (284.2 eV). The existence of nitrogen elements in the N, S-C/$In_2O_3$ DHR structure is mainly ascribed to the N-graphene (Fig. 3e), which is also supported by C 1s spectrum. The S 2p spectrum (Fig. 3f) can be deconvoluted into two peaks, assigning to S–C (167.3 eV and 168.4 eV). These results further confirm that the N, S-C/$In_2O_3$ DHR is comprised of $In_2O_3$ coated by the N,S co-doped carbon layer. The N and S co-doping could improve the electronic transport efficiency and chemical activity of the carbon layer, which could in turn enhance the separation efficiency of carriers in photocatalytic reactions.

**Photophysical properties of the photocatalysts.** The first step of photocatalysis is the generation of electron-hole pairs under light irradiation. Thus, the UV–vis absorption spectrum of the obtained N,S-C/$In_2O_3$ DHR was measured and compared with that of N,S-C/$In_2O_3$ NP ($In_2O_3$ nanoparticles coated by N,S-codoped carbon, Supplementary Fig. 8), N-C/$In_2O_3$ NP ($In_2O_3$ nanoparticles coated by N-doped carbon, Supplementary Fig. 9), $In_2O_3$ DHR (one-dimensional DHRs assembled by $In_2O_3$ nanoparticles without the carbon layer, Supplementary Fig. 10), $In_2O_3$ HD (three-dimensional double-shelled hollow dodecahedron assembled by $In_2O_3$ nanoparticles without the carbon layer, Supplementary Figure 11), and commercial $In_2O_3$ in order to investigate the influence of unique structural features of N,S-C/$In_2O_3$ DHR on the light absorption ability. Figure 4a shows that the N,S-C/$In_2O_3$ DHR possesses the strongest and broadest absorption in a visible-light region under the same conditions, and the optical absorption of other five samples follows the order of N,S-C/$In_2O_3$ NP > N-C/$In_2O_3$ NP > $In_2O_3$ DHR > $In_2O_3$ HD > commercial $In_2O_3$. This result indicates that N,S-codoped carbon layer and DHR structure play a synergistic role in enhancing the optical absorption ability of N,S-C/$In_2O_3$ DHR.

The separation and migration of electron-hole pairs is the second step of photocatalysis. In order to investigate this step, we measured the electrochemical impedance spectroscopy (EIS), photocurrent density, and linear sweep voltammetry (LSV) of N,S-C/$In_2O_3$ DHR, N,S-C/$In_2O_3$ NP, N-C/$In_2O_3$ NP, $In_2O_3$ DHR, $In_2O_3$ HD and commercial $In_2O_3$. The LSV curves (Fig. 4b) of these six samples indicate that the N,S-C/$In_2O_3$ DHR has the highest cathodic current density (CCD) for reducing $H_2O$ to $H_2$. Moreover, the CCD of N,S-C/$In_2O_3$ NP is higher than that of N-C/$In_2O_3$ NP. We then compared the LSV curves of the three samples without the coated carbon layer ($In_2O_3$ DHR, $In_2O_3$ HD and commercial $In_2O_3$), and found that the $In_2O_3$ DHR showed higher CCD than $In_2O_3$ HD and commercial $In_2O_3$. Thus, it can be concluded that the N,S-codoped carbon layer and one-dimensional hollow rod structure are both beneficial to accelerate the charge transfer. As shown in Fig. 4c, the order of photocurrent density among these six samples is N,S-C/$In_2O_3$ DHR > N,S-C/$In_2O_3$ NP > N-C/$In_2O_3$ NP > $In_2O_3$ DHR > $In_2O_3$ HD > commercial $In_2O_3$, indicating that the N,S-C/$In_2O_3$ DHR also has the highest separation efficiency of photogenerated carriers. Their photoluminescence (PL) emission spectra (Supplementary Fig. 12) were recorded to prove the high-resolution rate of photogenerated carriers. The emission peak at 498 nm was assigned to the band-band PL phenomenon. The intensity of the peaks shows the order of N,S-C/$In_2O_3$ DHR < N,S-C/$In_2O_3$ NP < N-C/$In_2O_3$ NP < $In_2O_3$ DHR < $In_2O_3$ HD < commercial $In_2O_3$. The emission intensity of N,S-C/$In_2O_3$ DHR nearly vanish, meaning that the N,S-codoped carbon layer acts as the electron acceptor to effectively prevent the recombination of photogenerated carriers. This result is well consistent with the photocurrent density study. Figure 4d shows the electrochemical impedance spectra of these six samples. The N,S-C/$In_2O_3$ DHR presents the smallest semicircular curve in the Nyquist plot, indicating its lowest charge-transfer

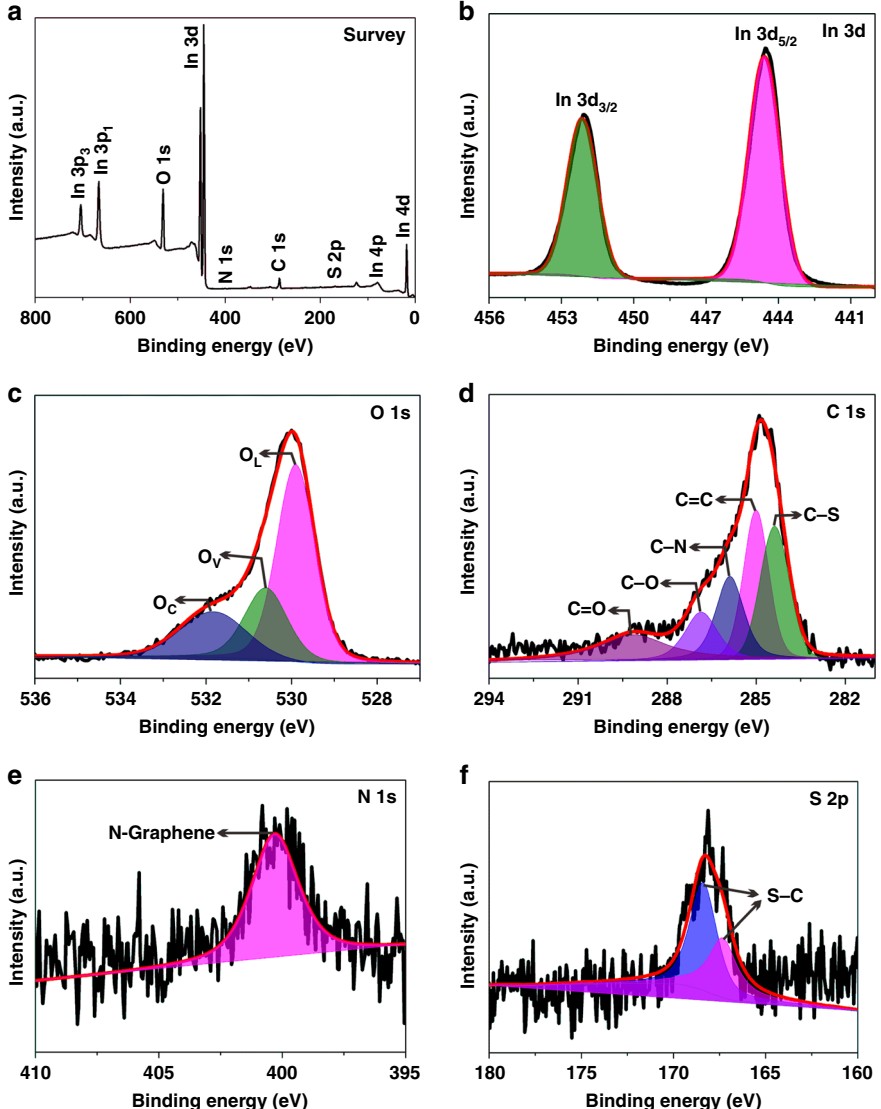

**Fig. 3** XPS spectra of N,S-C/In$_2$O$_3$ DHR. **a** Survey spectrum, and high-resolution spectra of **b** In 3d, **c** O 1s, **d** C 1s, **e** N 1s, and **f** S 2p. Source data are provided as a Source Data file

resistance, which is consistent with its more effective separation of photogenerated electrons and holes[47]. Therefore, N,S-codoped C layer evenly coated on In$_2$O$_3$ provides the electron storage and transfer channel to improve the separation efficiency of photogenerated carriers.

**Photocatalytic activity of the photocatalysts**. Photocatalytic activity of the as-prepared N,S-C/In$_2$O$_3$ DHR was evaluated by blue light LED induced oxidative hydroxylation of arylboronic acids (OHAA) to phenols. Phenols are very important intermediates for many polymers, natural products and pharmaceutical molecules[48–52]. The photocatalytic reactions were carried out using N,N-diisopropylethylamine (DIEA) as the sacrificing agent (Fig. 5a). The details for determining the optimized conditions for this photocatalytic reaction are shown in Supplementary Table 1. When compared with N,S-C/In$_2$O$_3$ NP, N-C/In$_2$O$_3$ NP, In$_2$O$_3$ DHR, In$_2$O$_3$ HD and commercial In$_2$O$_3$ (Fig. 5b and Supplementary Fig. 13), N,S-C/In$_2$O$_3$ DHR exhibits the highest catalytic activity with the production yield reaching about 99% after 12 h (determined by $^1$H NMR spectra, Supplementary Fig. 14). Compared to N-C/In$_2$O$_3$ HD (the In$_2$O$_3$ dodecahedron coated by N-doped C layer)[43], the catalytic time of N,S-C/In$_2$O$_3$ DHR for

achieving 99% production yield is reduced to half, indicating that the N,S-codoping plays an importantly positive role in improving the activity of carbon-coated photocatalysts. On the other hand, the photocatalyst is relatively less active under green and yellow light irradiation for 12 h, with the yields of 44 and 15%, respectively (Supplementary Fig. 15). The photocatalytic activity of other five samples follows the trend of N,S-C/In$_2$O$_3$ NP > N-C/In$_2$O$_3$ NP > In$_2$O$_3$ DHR > In$_2$O$_3$ HD > commercial In$_2$O$_3$, which is consistent with the analytic results shown in Fig. 4. Therefore, effective photocatalytic activity of N,S-C/In$_2$O$_3$ DHR should be assigned to the efficient separation of photogenerated carriers induced by uniform coating of the N,S-codoped C layer and enhanced optical absorption provided by the N,S-codoped C layer and DHR structure.

In order to access the recycling stability of N,S-C/In$_2$O$_3$ DHR, the same photocatalytic OHAA was carried out for five cycles with an interval of 12 h. The N,S-C/In$_2$O$_3$ DHR exhibits high recycling stability and stable photocatalytic efficiency (Fig. 5c). The shape and structure of the N,S-C/In$_2$O$_3$ DHR did not show obvious changes after photocatalytic cycle experiments (Supplementary Fig. 16). After keeping the solutions for 12 h, N,S-C/In$_2$O$_3$ DHR precipitated to the bottom of solution, while N,S-C/

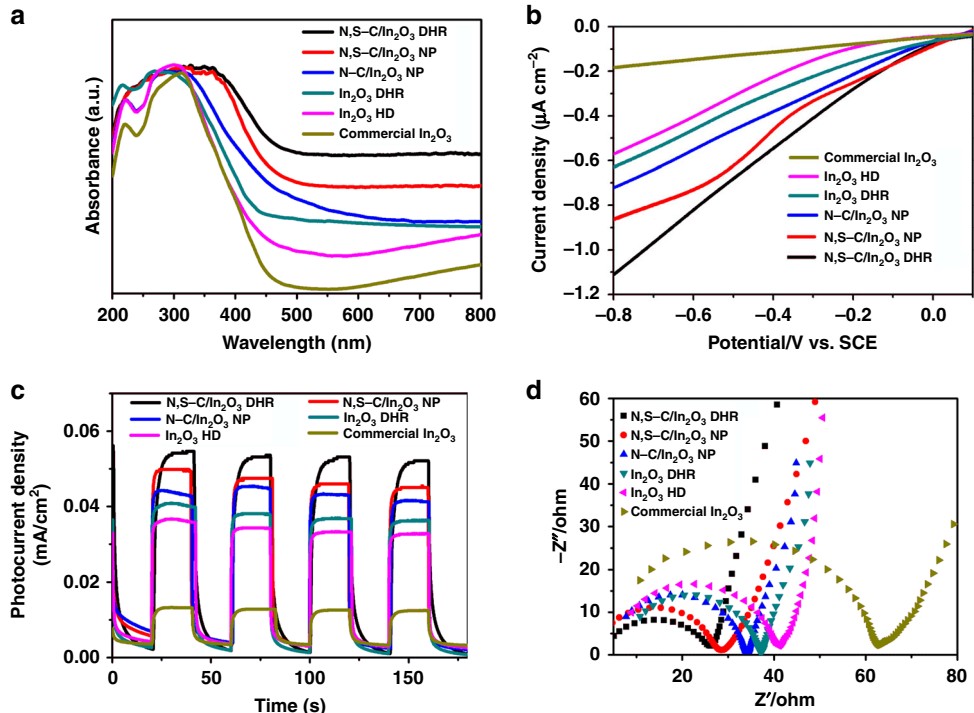

**Fig. 4** Optical absorption and photoelectric characterizations. **a** UV–vis absorption spectra, **b** LSV curves, **c** Photocurrent density measured at 0.2 V versus Hg/Hg$_2$Cl$_2$ under non-illuminated (i.e., in the dark) and illuminated conditions, and **d** EIS Nyquist plots for N,S-codoped In$_2$O$_3$ DHR, N,S-C/In$_2$O$_3$ NP, N-C/In$_2$O$_3$ NP, In$_2$O$_3$ DHR, In$_2$O$_3$ HD, and commercial In$_2$O$_3$. Source data are provided as a Source Data file

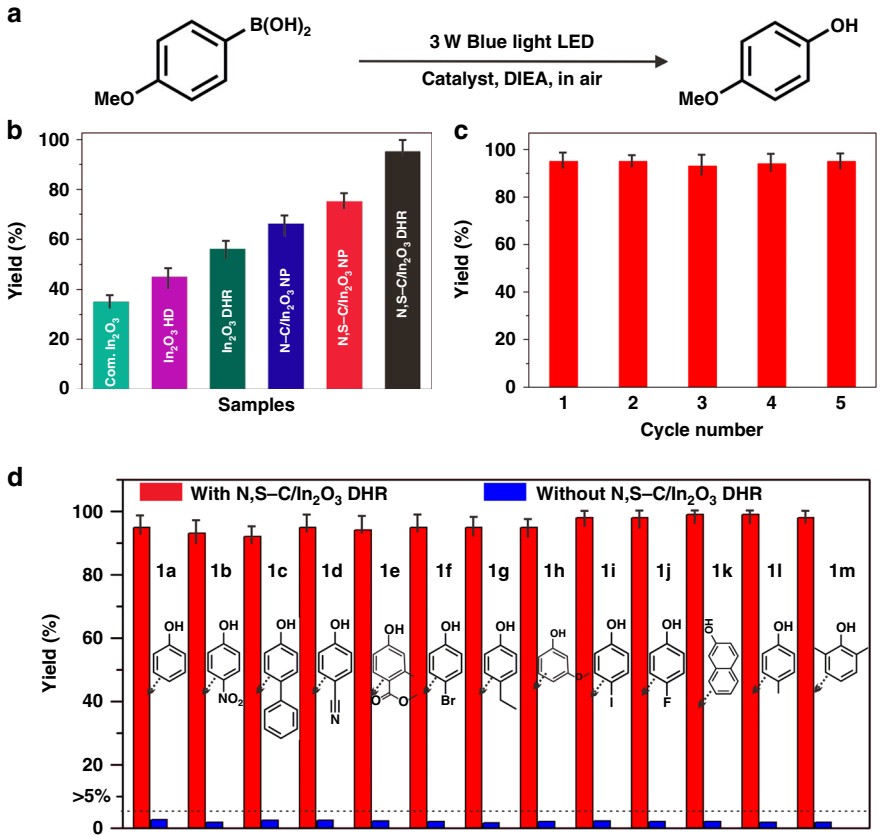

**Fig. 5** Photocatalytic oxidative hydroxylation. **a** Photocatalytic reaction for OHAA. **b** Reaction yields from OHAA using N,S-C/In$_2$O$_3$ DHR, N,S-C/In$_2$O$_3$ NP, N-C/In$_2$O$_3$ NP, In$_2$O$_3$ DHR, In$_2$O$_3$ HD, and commercial (com.) In$_2$O$_3$ as catalysts for 12 h. **c** Recycling tests for OHAA using N,S-C/In$_2$O$_3$ DHR as a catalyst under blue LED irradiation. **d** Reaction yields of various phenols obtained from oxidative hydroxylation of different arylboronic acids with and without N,S-C/In$_2$O$_3$ DHR as a photocatalyst. Source data are provided as a Source Data file

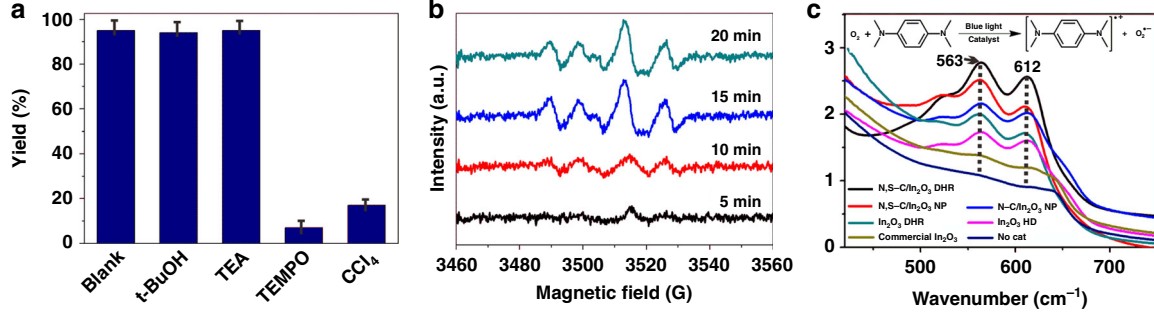

**Fig. 6** Photocatalytic mechanism studies. **a** Control experiments for OHAA over N,S-C/In$_2$O$_3$ DHR under blue LED irradiation in the presence of different scavengers: no scavenger (blank), t-BuOH, TEA, TEMPO, and CCl$_4$. **b** EPR spectra of a reaction solution in the present of DMPO under blue LED irradiation for different times. **c** UV–vis absorption spectra for cationic radicals of TMPD treated by N,S-C/In$_2$O$_3$ DHR, N,S-C/In$_2$O$_3$ NP, N-C/In$_2$O$_3$ NP, In$_2$O$_3$ DHR, In$_2$O$_3$ HD, and commercial In$_2$O$_3$ under blue LED light irradiation. Source data are provided as a Source Data file

In$_2$O$_3$ NP still suspended in the solution (Supplementary Fig. 17). Therefore, the N,S-C/In$_2$O$_3$ DHR can be recovered from the solution through simple centrifugation. We also measured the dispersity of N,S-C/In$_2$O$_3$ DHR in the reaction solution. N,S-C/In$_2$O$_3$ DHR can be well dispersed in the solution under continuous stirring (Supplementary Fig. 18). The catalytic activity of the N,S-C/In$_2$O$_3$ DHR for OHAA with various substrates was also investigated under the same catalytic experimental conditions (Fig. 5d). The corresponding reaction yields of the products are about 90–95%, indicating that the efficient photocatalytic activity of N,S-C/In$_2$O$_3$ DHR is not specific to only one reaction, but highly suitable for a series of OHAA.

**Photocatalytic mechanism for OHAA.** In order to understand the photocatalytic mechanism using N,S-C/In$_2$O$_3$ DHR as the photocatalyst, the energy band position of In$_2$O$_3$ was first calculated. As show in Supplementary Fig. 19, the absorption edge presents at 443 nm in visible-light region. The bandgap of the In$_2$O$_3$ DHR was estimated to 2.8 eV, matching well with the reported value[53]. Thus, the In$_2$O$_3$ DHR works well under blue LED light irradiation ($E = 2.76 \pm 4$ eV)[54]. The VB potential of In$_2$O$_3$ can be calculated by Mulliken electronegativity theory:

$$E_{VB} = X - E^e + 0.5E_g \qquad (1)$$

where $E_{VB}$ is corresponding VB potential, $X$ is the absolute electro-negativity of corresponding semiconductor, $E^e$ is the energy of free electrons in the hydrogen scale (about 4.5 eV), and $E_g$ is the bandgap energy of corresponding semiconductor. The $X$ value for In$_2$O$_3$ is 5.19 eV, and thus the $E_{VB}$ value of In$_2$O$_3$ is calculated to be 2.09 eV. The $E_{VB}$ value was also confirmed by the VB XPS (Supplementary Fig. 20). The CB potential ($E_{CB}$) can be obtained by:

$$E_{CB} = E_{VB} - E_g \qquad (2)$$

In order to reflect the effect of pH, the following equation was used to calculate the $E_{VB}$ and $E_{CB}$ potentials of the photocatalyst:

$$E = E^0 - 0.05915 \times pH \qquad (3)$$

Thus, the $E_{VB}$ and $E_{CB}$ values of In$_2$O$_3$ at pH = 7 were estimated to be 1.68 and $-1.12$ V vs normal hydrogen electrode (NHE), respectively.

A series of comparative experiments were performed by adding various radical scavengers into the reaction system to investigate the major reactive species in the photocatalytic OHAA. Tertiary butanol (t-BuOH) was adopted to extinguish hydroxyl radical (·OH), triethylamine (TEA) was introduced to quench photogenerated hole (h$^+$), (2,2,6,6-tetramethylpiperidin-1-yl)oxidanyl

(TEMPO) was employed to scavenge superoxide anion radical (·O$_2$$^-$), and CCl$_4$ was used for trapping photogenerated electron (e$^-$)[55–57]. As illustrated in Fig. 6a, before and after adding TEA and t-BuOH, the yield of corresponding phenol remains almost unchanged (Supplementary Figures 21–23). When TEMPO is added to quench ·O$_2$$^-$, the yield of phenol is significantly suppressed (Supplementary Figure 24). Noticeable inhibition can be also observed when CCl$_4$ is added to trap e$^-$ (Supplementary Figure 25). These results reveal that the e$^-$ and ·O$_2$$^-$ play the key roles in OHAA using N,S-C/In$_2$O$_3$ DHR as the photocatalyst. Electron paramagnetic resonance (EPR) was used to prove the existence of ·O$_2$$^-$, which was captured by adding 5,5-dimethyl-1-pyrroline-N-oxide (DMPO). Obvious characteristic signals of ·O$_2$$^-$ can be detected when DMPO is used as the radical scavenger, and the intensity of the signals gradually increases upon the illumination time (Fig. 6b). The results indicate that ·O$_2$$^-$ is the primary active intermediate species in this photocatalytic process.

The superoxide anion radical ·O$_2$$^-$ can only be produced by reacting dissolved O$_2$ with photogenerated electrons under the CB potential of O$_2$/·O$_2$$^-$ ($-0.33$ V vs NHE). In this system, the CB potential ($-1.12$ V vs NHE) of In$_2$O$_3$ is high enough to form ·O$_2$$^-$ based on the thermodynamic viewpoint. The generation of ·O$_2$$^-$ in the N,S-C/In$_2$O$_3$ DHR system was further verified by using N,N,N′,N′-tetramethyl-phenylenediamine (TMPD), because the photoactive systems can regulate the electrons transfer from TMPD molecule to O$_2$, resulting in the production of ·O$_2$$^-$ and blue-colored product with obvious absorption peaks at 612 nm and 563 nm[58]. All of the six samples (N,S-C/In$_2$O$_3$ DHR, N,S-C/In$_2$O$_3$ NP, N-C/In$_2$O$_3$ NP, In$_2$O$_3$ DHR, In$_2$O$_3$ HD and commercial In$_2$O$_3$) produce characteristic absorption peaks after reacting with TMPD (Fig. 6c), indicating that In$_2$O$_3$ has the ability to generate ·O$_2$$^-$. The characteristic absorption peak intensity of these six samples follows the order of N,S-C/In$_2$O$_3$ DHR > N,S-C/In$_2$O$_3$ NP > N-C/In$_2$O$_3$ NP > In$_2$O$_3$ DHR > In$_2$O$_3$ HD > commercial In$_2$O$_3$, which is consistent with the trend of the separation and migration ability of photogenerated carriers. The high-resolution O 1s spectra of these six In$_2$O$_3$ samples were measured to investigate the surface structures of photocatalysts (Supplementary Fig. 26 and Supplementary Table 2). Relative proportion of the oxygen vacancy (O$_V$) component in these samples is in the order of N,S-C/In$_2$O$_3$ DHR > N,S-C/In$_2$O$_3$ NP > N-C/In$_2$O$_3$ NP > In$_2$O$_3$ DHR > In$_2$O$_3$ HD > commercial In$_2$O$_3$, indicating that the O$_V$ provides the site for activating O$_2$ to ·O$_2$$^-$. In the N,S-C/In$_2$O$_3$ DHR system, ·O$_2$$^-$ is the main reactive species for photocatalytic OHAA, and e$^-$ is responsible for reducing the oxygen molecule to produce ·O$_2$$^-$.

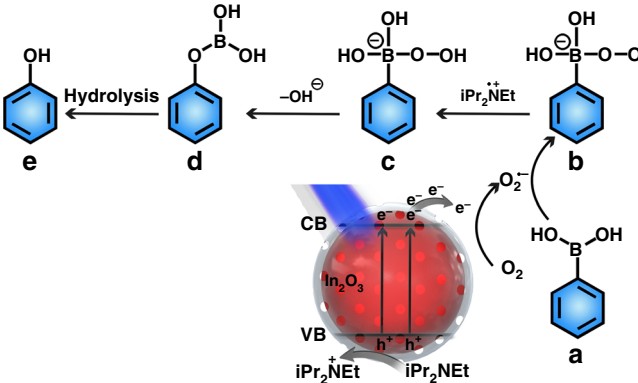

**Fig. 7** Schematic illustration exhibiting the proposed reaction mechanism (a–e) for OHAA using N,S-C/In₂O₃ DHR as the photocatalyst

Based on above studies, a possible photocatalytic mechanism of OHAA over N,S-C/In₂O₃ DHR was proposed (Fig. 7). Firstly, the photogenerated carriers are excited in N,S-C/In₂O₃ DHR under blue LED irradiation. Then, the photogenerated electrons are transferred to the N,S-codoped carbon layer, and the photogenerated holes still stay on the In₂O₃ nanoparticles. Secondly, the adsorbed O₂ molecule is reduced by separated electrons, resulting in the formations of ·O₂⁻. The DIEA molecule (iPr₂Net) can capture the separated holes to form iPr₂N⁺Et. Thirdly, the obtained ·O₂⁻ would react with arylboronic acid molecule (a) to generate intermediate (b). The intermediate (b) then abstracts a hydrogen atom from iPr₂N⁺Et to form intermediate (c), which is further rearranged with the loss of a –OH⁻ ion to form (d). The intermediate (d) is subsequently hydrolyzed to afford the final phenol product (e).

Microcosmic mechanism of photogenerated carrier separation and transfer on N,S-codoped In₂O₃ was further investigated by calculating the projected density of states (PDOS) from In₂O₃ (001) face/N,S-codoped graphite (001) face, In₂O₃ (110) face/N,S-codoped graphite (001) face, and In₂O₃ (111) face/N,S-codoped graphite (001) face, respectively. As shown in Fig. 8, the hybrid states of C 2p orbital, S 3p orbital, and N 2p orbital cross over the bandgap of In₂O₃ in the three In₂O₃/N,S-codoped graphite interfaces, meaning that the electrons from O 2p state of In₂O₃ would be transferred to these hybrid states of graphite. Moreover, the hybridization of N 2p, S 3p, and C 2p increases the strength of interstitial states to further improve the separation and transfer rate of photogenerated carriers. Thus, electrons are mainly excited from O 2p state of In₂O₃ to hybrid states of graphite (N 2p, S 3p, and C 2p orbitals) through the interface, leaving the photogenerated holes on O 2p state of In₂O₃.

## Discussion

In conclusion, the double-shelled hollow rods (N,S-C/In₂O₃ DHR) assembled by N,S-codoped C coated In₂O₃ ultrafine nanoparticles have been synthesized by thermolysis of In-based frameworks. The obtained N,S-C/In₂O₃ DHR possesses effective catalytic activity for photocatalytic OHAA with high stability and easy recovery at the same time. The excellent performance of N,S-C/In₂O₃ DHR in the photocatalysis is due to the following four structural characteristics. Firstly, the hollow rods not only provide a local microenvironment for OHAA, but also enhance the light absorption by increasing the optical reflection in the hollow interior to generate more electrons and holes. Secondly, the double shells of N,S-C/In₂O₃ DHR have lots of reactive sites, which improve the photocatalytic activity. Thirdly, the mesoporous structure of the shells improves the accessibility of N,S-C/In₂O₃ DHR, which is beneficial for the diffusion of reactants and

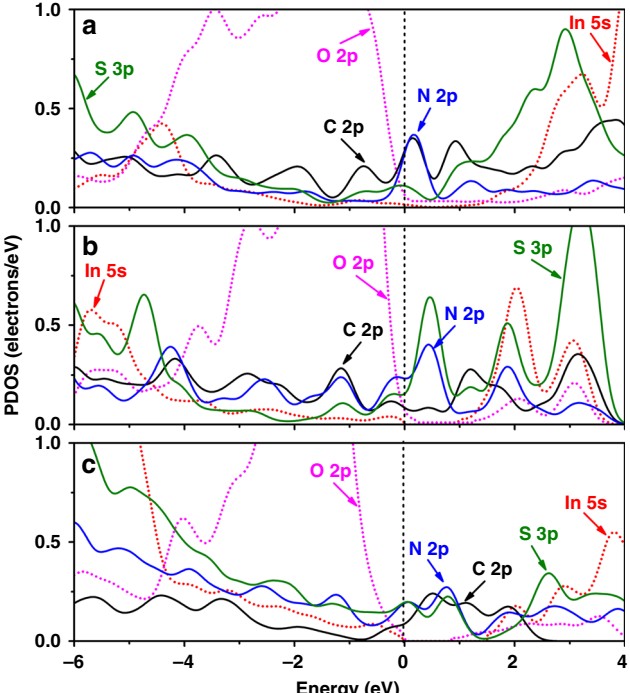

**Fig. 8** Density functional theory calculations. PDOS for **a** In₂O₃ (100) face/N,S-codoped graphite (001) face, **b** In₂O₃ (110) face/N,S-codoped graphite (001) face, and **c** In₂O₃ (111) face/N, S-codoped graphite (001) face

products. Fourthly, the regular morphology and large size (6–7 μm in length) of N,S-C/In₂O₃ DHR significantly improve the recyclability and reusability. The photocatalytic mechanism of OHAA over N,S-C/In₂O₃ DHR has been proposed, confirming that ·O₂⁻ is the primary reactive species in this catalytic reaction. In addition, the microcosmic separation and transfer pathways via the interface of In₂O₃/N,S-codoped graphite have been clarified by PDOS theoretical calculations, i.e., photogenerated electrons are mainly excited from O 2p state of In₂O₃ and transferred to hybrid states of the graphite consisting of S 3p, N 2p, and C 2p orbitals, leaving photogenerated holes on O 2p state of In₂O₃. Thus, this work presents useful information on the rational exploration of In₂O₃-based hybrid materials for achieving highly efficient photocatalysts.

## Methods

**Synthesis of N,S-codoped MIL-68-In**. Indium(III) nitrate hydrate (0.21 mmol, 0.08 g), 1,4-benzenedicarboxylic acid (0.18 mmol, 0.03 g) and BIT (0.13 mmol, 0.02 g) were separately added into N,N-dimethylformamide (10 mL). After ultrasonication for 10 min, the above solution was directly transferred to a Teflon-lined steel autoclave (50 mL) and put in oven at 100 °C under static conditions for 4 h. When the reaction completed, the sample was obtained by centrifugation, washed with ethanol for several times, and finally dried at 60 °C for 24 h.

**Synthesis of N,S-C/In₂O₃ DHR**. N,S-C/In₂O₃ DHR was obtained via the pyrolysis of the N,S-codoped MIL-68-In precursor in a vacuum atmosphere at 550 °C with the heating rate of 5 °C min⁻¹ for 45 min.

**Synthesis of In₂O₃ DHR**. In₂O₃ DHR was synthesized via the pyrolysis of the obtained N,S-C/In₂O₃ DHR in O₂ at 550 °C with the heating rate of 5 °C min⁻¹ for 45 min.

**Synthesis of N,S-C/In₂O₃ NP**. Indium(III) nitrate hydrate (0.21 mmol, 0.08 g), benzimidazole (0.25 mmol, 0.03 g), 1,4-benzenedicarboxylic acid (0.18 mmol, 0.03 g) and BIT (0.13 mmol, 0.02 g) were dissolved in methanol (10 mL). After ultrasonication for about 10 min, the solution was directly poured into a Teflon-lined steel autoclave (50 mL), and put in an oven at 100 °C under static conditions for 240 min. When the reaction finished, the product was obtained through the centrifugation, which was washed several times with ethanol and finally dried at

60 °C for 24 h. N,S-C/$In_2O_3$ NP was obtained by the pyrolysis of the product in a vacuum environment at 450 °C with the heating rate of 2 °C min$^{-1}$ for 1 h.

**Synthesis of N-C/$In_2O_3$ NP**. Indium(III) nitrate hydrate (0.21 mmol, 0.08 g), benzimidazole (0.254 mmol, 0.03 g) and 1,4-benzenedicarboxylic acid (0.18 mmol, 0.03 g) were separately added into methanol (10 mL). After ultrasonication for about 10 min, the solution was directly poured into a Teflon-lined steel autoclave (50 mL), and put in the oven at 100 °C under static conditions for 240 min. When the reaction finished, the product was obtained by centrifugation, washed with ethanol for several times, and finally dried at 60 °C for 24 h. N-C/$In_2O_3$ NP was obtained by the pyrolysis of the product in a vacuum environment at 450 °C with the temperature increasing speed of 2 °C min$^{-1}$ for 1 h.

**Synthesis of $In_2O_3$ HD**. Indium(III) nitrate hydrate (0.040 mmol, 0.015 g), benzimidazole (0.846 mmol, 0.1 g) and 4,5-imidazoledicarboxylic acid (0.135 mmol, 0.021 g) were separately dissolved in N,N-dimethylformamide (6 mL). After ultrasonication for about 10 min, the solution was directly transferred to a round bottom flask (50 mL) heated at 100 °C for 4 h. When the reaction finished, the sample was obtained by centrifugation, washed with ethanol for several times, and then dried at 60 °C for 24 h. The pyrolysis was performed on the sample in an argon environment at 500 °C with the heating rate about 2 °C min$^{-1}$ for 1 h to afford product A. $In_2O_3$ HD was obtained via the pyrolysis of the product A in an oxygen environment at 500 °C with a heating rate about 2 °C min$^{-1}$ for 6 h.

**Characterizations**. The compositions and phases of the obtained samples were characterized by powder XRD on the PANalytical X'Pert diffractometer with a CuKα radiation. SEM (SU8100) and HRTEM (FEI Tecnai-F20) were used to characterize the shape and crystal structure. The surface compositions of samples were characterized by PHI QUANTUM2000 photoelectron spectrometer (XPS). $N_2$ adsorption/desorption isotherms were used to characterize the surface areas of samples based on the BET method (Micrometics ASAP 2020 system). Bruker ESP-300E spectrometer was used to measure EPR spectra at 9.8 GHz with X-band and 100 Hz field modulation. Dynamic light scattering (DLS) experiments were conducted on a Nano-Zetasizer (Nano-ZS) from Malvern Instruments (Malvern, UK). The experimental process was as follows: samples (0.1 g) and ethanol (10 mL) were added into a glass bottle (15 mL) and dispersed uniformly by ultrasonication for 10 min. Solutions were placed at different times under agitation and non-agitation, respectively. Then, the upper liquids were taken for DLS tests.

**Photoelectrochemical measurements**. Photoelectrochemical experiments were measured in the three electrode quartz cell. The Pt plate was selected as the counter electrode, the reference electrode was Hg/HgCl$_2$ electrode, and the corresponding working electrode was obtained on the fluorine doped tin oxide (FTO) glass. In order to obtain a slurry, the obtained product (10 mg) was ultrasonicated in ethanol (0.2 mL). The slurry was then spread onto the FTO glass with the other side protected by using Scotch tape. After drying in air, the Scotch tape was unstuck. In order to improve the adhesion, the obtained working electrode with the area about 2.5 cm$^2$ was dried at 300 °C for 180 min. The CHI-760E workstation was used to measure the EIS. In the three-electrode cell, the 0.025 M $KH_2PO_4$ and 0.025 M $Na_2HPO_4$ standard buffer solution (25 °C, pH = 6.864) were selected as the electrolytes without adding any additive, and the measurements were performed on an open circuit potential condition. A 300 W Xe arc lamp system was used as the visible-light irradiation source. The LSV technique was employed to obtain cathodic polarization curves.

**Photocatalytic reactions**. The photocatalyst (5 mg), DIEA (0.3 mmol), and phenylboronic acid (0.1 mmol, 1 equivalent) were separately dissolved into ethanol (2 mL). The stirring solution was irradiated under the blue LED (3 W, 450 nm) at room temperature in air. The yield of the product was characterized by $^1H$ NMR spectra. To perform the recycling experiments, the photocatalyst was recovered by centrifugation and washed with dichloromethane for several times. The recycled photocatalyst was then dried in vacuum at about 60 °C for 24 h.

**Density functional calculations**. Plane-wave pseudopotential method was used to calculate the density functional, which was implemented in the Cambridge Sequential Total Energy Package (CASTEP) code[59]. The exchange-correlation effects and electron-ion interactions were described by the local density approximation[60] and ultrasoft pseudopotential[61], respectively. A Morkhost-Pack mesh[62] of Γ and 2 × 2 × 1 point were used to calculated geometry optimization and electronic properties, respectively. The self-consistent convergence accuracy, the convergence criterion for the force between atoms and the maximum displacement were set at 2 × 10$^{-6}$ eV/atom, 5.5 × 10$^{-2}$ eV/Å, and 2 × 10$^{-3}$ Å, respectively.

**Reporting summary**. Further information on research design is available in the Nature Research Reporting Summary linked to this article.

## Data availability

The authors declare that the main data supporting the findings of this study are available within the paper and its Supplementary Information. The source data underlying Figs. 2, 3, 4, 5b–d and 6a–c and Supplementary Figs. 1a,i,q, 2a,b, 4, 5, 7–16, and 19–25 are provided as a Source Data file.

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

## Acknowledgements

This work was supported by the National Natural Science Foundation of China (21671085 and 21701063), the Jiangsu Province Science Foundation for Youths (BK20150237), the Natural Science Foundation of Jiangsu Province (BK20161160), the Qing Lan Project, and the Project Funded by the Priority Academic Program Development of Jiangsu Higher Education Institutions. It was also supported by the Singapore Agency for Science, Technology and Research (A*STAR) AME IRG grant (A1783c0007).

## Author contributions

X.H. and Y.Z. conceived and designed the experiments and wrote the paper. L.S., R.L., and Y.Y. performed most of experiments and theoretical calculation. X.W. and W.Z. guided photocatalytic experiments. The manuscript was written through contributions of all authors. All authors have given approval to the final version of the manuscript.

## Additional information

**Competing interests:** The authors declare no competing interests.

