## [Peer Review File · Nature Communications]

Reviewers' comments:

Reviewer #1 (Remarks to the Author):

In this manuscript, the authors reported a synthetic method of double-shelled hollow rods assembled by N,S-codoped carbon coated In₂O₃ ultrafine nanoparticles (N,S-C/In₂O₃ DHR). The characterization (XRD, SEM, TEM, mapping and XPS) of the prepared NCs was performed clearly, which provided significant scientific insights. A series of contrast experiments provided a thorough investigation of photoelectric properties of N,S-C/In₂O₃ DHR. By contrast with different catalysts, N,S-C/In₂O₃ DHR exhibited excellent photocatalytic activity toward photocatalytic oxidative hydroxylation of various arylboronic acid substrates. Moreover, authors investigated the photocatalytic mechanism by using UV-vis absorption spectra, ESR spectra and theoretical calculation. As we known, it is challenging to synthesize a photocatalyst combining with the functions of high catalytic activity, high stability and easy recovery. In this article, authors achieved the above goals by designing the structure and composition of catalyst. Therefore, it is an interesting and innovative work in the field of the controllable syntheses of metallic oxide photocatalysts. The manuscript seems also well-organized. Therefore, I am glad to recommend the publication of the manuscript in Nature Communications after the revision of following minor issues:

1. In the photocatalytic oxidative hydroxylation of arylboronic acids, authors used blue LED light irradiation, how about the yield when irradiated with green and yellow LED light?
2. Arylboronic acids substrate scope should be further expanded.
3. In Scheme 2, what is the origin of oxygen atom in the phenol product (Ar-O-H)? Is it from that in O₂ molecules?
4. In the photocatalytic experiment, ¹H NMR was employed to determine the yield. The ¹H NMR spectra of reactant and product should be provided.
5. XPS spectra determined the chemical states of each element in the material. Therefore, the positions of each sub-peaks indicated by XPS spectra should be shown more clearly in other ways.
6. In order to measure the recovery of N,S-C/In₂O₃ DHR, authors compared the precipitation of N,S-C/In₂O₃ DHR and N,S-C/In₂O₃ NP after keeping their solutions for 12h, and the corresponding results were shown in Fig. S14. But the sample in Fig. S14 was white, which was not the color for C coated sample. Authors should explain this phenomenon.
7. The VB potential of In₂O₃ was calculated to be 2.09 eV. Authors should confirmed this value by VB XPS.

Reviewer #2 (Remarks to the Author):

Han, Zhao and coworkers developed the double-shelled hollow rods assembled from N,S-codoped carbon coated In₂O₃ nanoparticles (N,S-C/In₂O₃ DHR) and their application as photocatalysts, which is of fundamental importance in photocatalysis and will abstract broad interests. The photoelectric performances of N,S-C/In₂O₃ DHR are well characterized. Thus, we recommended to publish this manuscript in Nature Communications after minor revisions. The comments and arguments are listed as shown below:

1) According to the gradient variance tendency shown in Fig. 3 by comparing the N,S-codoped In₂O₃ DHR, N,S-C/In₂O₃ NP, N-C/In₂O₃ NP, In₂O₃ DHR, In₂O₃ HD and commercial In₂O₃, it seems that the In₂O₃-based materials with higher crystal integrities exhibit higher electronic resistances, under the ground or excited states. Also, based upon the statement of photocatalytic mechanism, the oxygen of In₂O₃ donated an electron to O₂ upon photo-irradiation. Thus, it was deduced that, the defects of In₂O₃, especially that of oxygen sites in In₂O₃, plays an important role in determining the photoelectronic performances. It was suggested to check the high-resolution XPS spectrum of those In₂O₃-based materials and compare the different compositions of O1s, for the rationalization of structure-activity relationship.

2) In the photocatalytic section, the author stated that the N,N-diisopropylethylamine was a “co-catalyst”, but this is not correct, it should be the hydrogen and electron donor or the so called sacrifice agent.

Reviewer #3 (Remarks to the Author):

The manuscript by Sun et. al reports a method to obtain N,S-C/In₂O₃ photocatalysts with the morphology of double-shelled hollow rods. In general, this is a well-organized paper containing interesting results and the characterizations of the materials are sufficient. The key of this method is adding BIT as the modulator and the N, S sources while synthesis In-based MOF. This seems a very simple and effective method for uniformly doping heteroatoms to MIL-68-In. However, the fabrications of MOF-derived catalytic materials have been well know. It also seems difficult to extend the reported strategy to other systems, raising questions on its universality. I did not find any conceptual breakthrough that would greatly advance the field. I was not convinced this work is of broad interest to the general research community or its meets the high standards of Nature Communications in terms of novelty, significance and impact. Overall, this is an interesting piece of work but a more specialized journal is suggested. A few comments are provided below.

1. The authors might want to emphasize the novelty and significance of the work. Particularly, how good is the catalytic performance by comparing with reported system?
2. The authors mentioned that their catalysts could easily be recovered. However, this means the catalysts have very poor colloidal dispersity. Undoubtably, this should not be called as an advantage! If the catalysts cannot be well dispersed, how good can their catalytic activity be?
3. I was trying to find the challenge that the current work aimed to address. In the abstract, it was claimed “Excellent catalytic activity, high stability and easy recovery are three key elements for fabricating an efficient photocatalyst”. However, this claim is too vague for judging the importance and difficulty of the challenge. What is the exact challenge that is addressed in this work? Why is it important? Why is still not addressed in previous studies?
4. When different amounts of competitive or modulator ligand were involved, the morphology can be controlled (J. Am. Chem. Soc. 2011, 133, 15506–15513; Angew. Chem. 2009, 121, 4833 –4837). Would the shape change with different amounts of BIT? If not, the effect of doping amount should be performed.
5. Some information is lack of consistency. e.g. “Hence, annealing N,S-codoped MIL-68-In at 500 oC in a vacuum atmosphere with a ramping rate of 2 °C min⁻¹ was sufficient to ensure complete conversion from MIL-68-In to In₂O₃.” (line 109-111 in the main text) and “N,S-C/In₂O₃ DHR was

synthesized via the calcination of the obtained N,S-codoped MIL-68-In at 550 °C in vacuum atmosphere with heating rate of 5 °C min⁻¹ for 45 min." (line 13-14 in the supporting information)
What are the exact calcination temperature and heating rate?

Response to Reviewer #1's Comments:

Comments to the Author:

In this manuscript, the authors reported a synthetic method of double-shelled hollow rods assembled by N,S-codoped carbon coated In_2O_3 ultrafine nanoparticles (N,S-C/ In_2O_3 DHR). The characterization (XRD, SEM, TEM, mapping and XPS) of the prepared NCs was performed clearly, which provided significant scientific insights. A series of contrast experiments provided a thorough investigation of photoelectric properties of N,S-C/ In_2O_3 DHR. By contrast with different catalysts, N,S-C/ In_2O_3 DHR exhibited excellent photocatalytic activity toward photocatalytic oxidative hydroxylation of various arylboronic acid substrates. Moreover, authors investigated the photocatalytic mechanism by using UV-vis absorption spectra, ESR spectra and theoretical calculation. As we known, it is challenging to synthesize a photocatalyst combining with the functions of high catalytic activity, high stability and easy recovery. In this article, authors achieved the above goals by designing the structure and composition of catalyst. Therefore, it is an interesting and innovative work in the field of the controllable syntheses of metallic oxide photocatalysts. The manuscript seems also well-organized. Therefore, I am glad to recommend the publication of the manuscript in Nature Communications after the revision of following minor issues:

Response: We really appreciate the reviewer's useful comments and positive recommendation of publication after revisions. We have carefully revised the manuscript based on the comments.

1. In the photocatalytic oxidative hydroxylation of arylboronic acids, authors used blue LED light irradiation, how about the yield when irradiated with green and yellow LED light?

Response: Thanks for the interesting suggestion. According to the advice, the photocatalytic activity of N,S-C/ In_2O_3 DHR toward oxidative hydroxylation of arylboronic acids was evaluated by using green and yellow LED light. On account of weak absorption of N,S-C/ In_2O_3 DHR to green and yellow light, the yield was relatively lower, *i.e.*, 44% under green light and 15% under yellow light for 12 h irradiation (Figure R1). In order to have more detailed descriptions about the catalytic activity of N,S-C/ In_2O_3 DHR, Figure R1 had been added in the revised supporting file as Figure S15, and corresponding descriptions have been added in the revised manuscript (page 13).

Figure R1. Yields from oxidative hydroxylation of arylboronic acids using N,S-C/In₂O₃ DHR under different LED light irradiation.

2. Arylboronic acids substrate scope should be further expanded.

Response: Thanks for the kind reminder. According to the suggestion, we have further expanded the substrate scope, and corresponding experimental results (Figure R2) had been added as Figure 4d in the revised manuscript.

Figure R2. Reaction yields of various phenols obtained from oxidative hydroxylation of different arylboronic acids with and without N,S-C/In₂O₃ DHR as a catalyst.

3. In Scheme 2, what is the origin of oxygen atom in the phenol product (Ar-O-H)? Is it from that in O₂ molecules?

Response: Thanks a lot for the question. Yes, it is from O₂ molecule. We used electron spin resonance to capture superoxide anion radical ($\cdot\text{O}_2^-$), an important intermediate formed by reacting oxygen with photo-generated electrons for the photocatalysis (please refer to pages 17-18). Other studies also indicated that the oxygen atom in the phenyl product originates from O₂ molecule in photocatalytic oxidative hydroxylation (*Angew. Chem. Int. Ed.* **2012**, *51*, 784).

4. In the photocatalytic experiment, ^1H NMR was employed to determine the yield. The ^1H NMR spectra of reactant and product should be provided.

Response: Thanks for the kind reminder. According to the suggestion, ^1H NMR spectra (Figure R3) of both reactant and product have been added in the revised supporting file as Figure S14.

Figure R3. (a) ^1H NMR spectrum of reactant, and (b) ^1H NMR spectrum of product.

5. XPS spectra determined the chemical states of each element in the material. Therefore, the positions of each sub-peaks indicated by XPS spectra should be shown more clearly in other ways.

Response: Thanks for the kind suggestion. In order to more clearly show the positions of each sub-peak, we have added corresponding color to each sub-peak in the revised spectra (Figure 2).

6. In order to measure the recovery of N,S-C/ In_2O_3 DHR, authors compared the precipitation of N,S-C/ In_2O_3 DHR and N,S-C/ In_2O_3 NP after keeping their solutions for 12h, and the corresponding results were shown in Fig. S14. But the sample in Fig. S14 was white, which was not the color for C coated sample. Authors should explain this phenomenon.

Response: Thanks for the kind notice. We are sorry that we have described samples in original Figure S14 by the mistake. In fact, original Figure S14 shows the precipitation of N,S-C/ In_2O_3 DHR and commercial In_2O_3 . The color of commercial In_2O_3 is white. Original Figure S14 has been changed to Figure S17 (Figure R4). The legend of Figure S17 and corresponding descriptions have been corrected in the revised manuscript (page 15).

Figure R4. Sedimentation of (a) commercial In_2O_3 NP and (b) N,S-C/ In_2O_3 DHR for different time without agitation.

7. The VB potential of In_2O_3 was calculated to be 2.09 eV. Authors should confirmed this value by VB XPS.

Response: Thanks for the kind suggestion. The valence band X-ray photoelectron spectroscopy (VB XPS) of pure In_2O_3 is shown in Figure R5. It can be seen that the position of the valence band edge of In_2O_3 is located at about 2.15 eV, which is consistent with the conclusion calculated from the Mulliken electronegativity theory. Figure R5 has been added in the revised supporting file as Figure S20, and corresponding descriptions have been added in the revised manuscript (page 16).

Figure R5. VB XPS spectrum of pure In_2O_3 .

Response to Reviewer #2's Comments:

Comments to the Author:

Han, Zhao and coworkers developed the double-shelled hollow rods assembled from N,S-codoped carbon coated In_2O_3 nanoparticles (N,S-C/ In_2O_3 DHR) and their application as photocatalysts, which is of fundamental importance in photocatalysis and will abstract broad interests. The photoelectronic performances of N,S-C/ In_2O_3 DHR are well characterized. Thus, we recommended to publish this manuscript in Nature Communications after minor revisions. The comments and arguments are listed as shown below:

Response: We really appreciate the reviewer's useful comments and positive recommendation of publication after minor revisions. We have carefully revised the manuscript based on the comments.

1) According to the gradient variance tendency shown in Fig. 3 by comparing the N,S-codoped In_2O_3 DHR, N,S-C/ In_2O_3 NP, N-C/ In_2O_3 NP, In_2O_3 DHR, In_2O_3 HD and commercial In_2O_3 , it seems that the In_2O_3 -based materials with higher crystal integrities exhibit higher electronic resistances, under the ground or excited states. Also, based upon the statement of photocatalytic mechanism, the oxygen of In_2O_3 donated an electron to O_2 upon photo-irradiation. Thus, it was deduced that, the defects of In_2O_3 , especially that of oxygen sites in In_2O_3 , plays an important role in determining the photoelectronic performances. It was suggested to check the high-resolution XPS spectrum of those In_2O_3 -based materials and compare the different compositions of O1s, for the rationalization of structure-activity relationship.

Response: Thanks very much for the useful suggestion. According to the suggestion, we have checked the high-resolution O 1s spectra of these In_2O_3 -based materials. As shown in Figure R6, high resolution O1s XPS peaks of these samples had some slight differences. The O1s core level peaks could be resolved into three centered Gaussian components (*Appl. Surf. Sci.* **2000**, 158, 134). The O_L component (529.7 ± 0.2 eV) was assigned to the lattice oxygen in the In_2O_3 phase. The O_V peak at the medium binding energy (530.6 ± 0.2 eV) was usually corresponded to the O^{2-} ion in oxygen-deficiency region, and the O_c component (531.8 ± 0.2 eV) was associated with the dissociated and chemisorbed oxygen species (O^- , O_2^- , and O^{2-}). As shown in Table R1, the relative proportion of the O_V component in different samples is N,S-C/ In_2O_3 DHR > N,S-C/ In_2O_3 NP > N-C/ In_2O_3 NP > In_2O_3 DHR > In_2O_3 HD > commercial In_2O_3 , which is well consistent with the photocatalytic activity. The results indicated that the surface structure (especially oxygen deficiency sites of materials) was indeed an important factor affecting the catalytic performance of corresponding materials. This is also a very important topic to understand how surface oxygen defects could affect photocatalytic performance. Figure R6 and Table R1 have been added in the revised supporting file as Figure S26 and Table S2, respectively. The corresponding descriptions have

been added in the revised manuscript (pages 18-19).

Figure R6. XPS spectra and curve fitting of O 1s in different samples.

Table R1. Results of curve fitting of O 1s spectra in six different In_2O_3 samples.

Samples	O_L (In-O)	O_V (Vacancy)	O_C (Chemisorbed)
Commercial In_2O_3	529.6 55.5 %	530.4 20.6 %	531.7 23.9 %
In_2O_3 HD	529.6 46.5 %	530.6 24.4 %	531.8 29.1 %
In_2O_3 DHR	529.9 52.7 %	530.6 26.8 %	531.8 20.5 %
N-C/ In_2O_3 NP	529.8 49.7%	530.7 29.2%	531.9 21.1%
N,S-C/ In_2O_3 NP	529.6 56.4%	530.6 32.4%	531.9 11.2%
N,S-C/ In_2O_3 DHR	529.7 39.2%	530.7 41.9%	531.9 18.8%

2) In the photocatalytic section, the author stated that the N,N-diisopropylethylamine was a “co-catalyst”, but this is not correct, it should be the hydrogen and electron donor or the so called sacrifice agent.

Response: Thanks for the kind reminder. The description about “co-catalyst” has been changed to “sacrifice agent” in the revised manuscript (page 13).

Response to Reviewer #3's Comments:

Comments to the Author:

The manuscript by Sun et. al reports a method to obtain N,S-C/In₂O₃ photocatalysts with the morphology of double-shelled hollow rods. In general, this is a well-organized paper containing interesting results and the characterizations of the materials are sufficient. The key of this method is adding BIT as the modulator and the N, S sources while synthesis In-based MOF. This seems a very simple and effective method for uniformly doping heteroatoms to MIL-68-In. However, the fabrications of MOF-derived catalytic materials have been well know. It also seems difficult to extend the reported strategy to other systems, raising questions on its universality. I did not find any conceptual breakthrough that would greatly advance the field. I was not convinced this work is of broad interest to the general research community or its meets the high standards of Nature Communications in terms of novelty, significance and impact. Overall, this is an interesting piece of work but a more specialized journal is suggested. A few comments are provided below.

Response: Thanks for reviewing our manuscript. We really appreciate the reviewer's comments that the main point of this article is to develop a simple and effective method for uniformly doping heteroatoms to MIL-68-In by adding 1,2-benzisothiazolin-3-one (BIT) as the modulator and N,S sources. The novel double-shelled hollow rods were then assembled from N,S-codoped carbon coated In₂O₃ nanoparticles produced by calcining the N,S-codoped MIL-68-In. Although the fabrications of MOF-derived catalytic materials have been reported, using MOFs for the preparation of heteroatom-codoped carbon coated metal oxide hollow rods as excellent photocatalysts has not been well studied. [Redacted]

Excellent photocatalysts would have three of following qualities at the same time: the enhanced optical absorption abilities to increase the number of photo-generated charge carriers, the improved separation of photo-generated charge carriers to prolong the lifetime of carriers, and the large surface area to provide more reactive sites for photocatalytic reactions. One effective approach to achieve these qualities is to prepare metal oxide nanoparticles coated by a carbon layer as efficient photocatalysts. On account of excellent mobility of charge carriers, the carbon layer can act as photo-generated-electron acceptor to improve the separation of photo-generated electrons and holes. Increasing the specific surface area of photocatalysts by decreasing their sizes could provide more reactive sites. There are two bottlenecks that restrict the application of this approach in fabricating efficient photocatalysts. (1) It is difficult to control the coating of carbon layer and introduce the heteroatoms. In general, enough and intimate contact interface is the key factor ensuring the efficient transfer of photo-generated carriers. While metal oxide nanoparticles are generally loaded on carbon materials, only a small fraction of the nanoparticle surface is in direct contact with the carbon materials. The core-shell structures could provide a three-dimensional (3D) intimate contact and maximize the interface between carbon layers and metal oxide nanoparticles. But, the

uniformly coating carbon layer on metal oxide nanoparticles is still a challenging task. Recent studies reported that heteroatom-doped carbon could further improve the conductivity of carbon materials (*Adv. Mater.* **2015**, *27*, 6021-6028; *Nano Energy* **2016**, *19*, 373-381; *J. Mater. Chem. A* **2017**, *5*, 22964-22969). The traditional techniques for introducing heteroatoms into carbon materials have unavoidable disadvantages, limiting their practical applications, which include toxic precursors, special and/or sophisticated instruments, harsh conditions, and uneven doping atom distribution (*Nano Lett.* **2011**, *11*, 2472-2477; *J. Am. Chem. Soc.* **2012**, *134*, 11060-11063; *Adv. Funct. Mater.* **2016**, *26*, 5708-5717; *Carbon* **2017**, *118*, 511-516). Realizing the co-doping of double heteroatoms is even harder. Therefore, it is desirable but challenging to fabricate the ultrafine metal oxide nanoparticles coated by double heteroatoms along with co-doped carbon layers. (2) It is also difficult to recycle the ultrafine photocatalyst nanoparticles from solution. Though the ultrafine nanoparticles display excellent photocatalytic performance, the difficulty in recycling ultrafine nanoparticles may increase the cost of photocatalytic reactions and result in secondary pollution of water. While magnetic photocatalysts may facilitate the recycling of photocatalysts to some extent, the reaction conditions of magnetic photocatalysts might be limited, because the dispersion of magnetic photocatalysts in solution can only use mechanical stirring and cannot use magnetic stirring. Therefore, a rational design of ultrafine nanoparticles with high reactivity and easy recyclability is highly desirable for extending their photocatalytic application.

To date, only a few studies could overcome abovementioned two bottlenecks using one photocatalytic material system. In our study, we developed a simple and efficient method for fabricating the double-shelled hollow rods assembled by N,S-codoped carbon coated In₂O₃ ultrafine nanoparticles (N,S-C/In₂O₃ DHR) using MIL-68-In MOF as the template and BIT as the modulator in one step. The obtained N,S-C/In₂O₃ DHR possesses both uniform N,S-codoped carbon layers and easy recyclability from solution through simple centrifugation. In addition, the double-shelled hollow structure of N,S-C/In₂O₃ DHR not only improves light harvesting by simultaneously increasing the reflection on the outer shell and the diffraction on the hollow cavity to generate more electrons and holes, but also provides more active sites for photocatalytic reactions. Therefore, we have achieved very high photocatalytic performance by using N,S-C/In₂O₃ DHR. The present research puts forward a useful protocol for fabricating efficient photocatalysts, exhibiting its novelty and significance to warrant the publication in *Nature Communications*.

[Redacted]

1. The authors might want to emphasize the novelty and significance of the work. Particularly, how good is the catalytic performance by comparing with reported system?

Response: Thanks for the suggestion. We have addressed the novelty and significance of our work in the response to the general comments above. Here, we would like to further discuss

about it. Light absorption efficacy is still the primary factor limiting the practical applications of photocatalysis. Therefore, improving the sensitivity of photocatalysts to light is the key to the practical applications of photocatalysis. In literature, high-power Xe lamp and LED lamp were used as the photocatalytic light sources, such as 300W Xe lamp (*Angew. Chem. Int. Ed.* **2016**, *55*, 4697-4700; *Appl. Catal. B* **2019**, *243*, 10-18), 35W Xe lamp (*Chem. Commun.* **2013**, *49*, 8689-8691), and 34W blue LED lamp (*Science* **2016**, *352*, 1304-1308; *Science* **2016**, *353*, 279-283; *Nature* **2015**, *524*, 330-334; *J. Am. Chem. Soc.* **2016**, *138*, 8084-8087; *J. Am. Chem. Soc.* **2016**, *138*, 12719-12722). In our work, we used 3W blue LED lamp as the photocatalytic light source, indicating that the as-prepared N,S-C/In₂O₃ DHR possesses highly improved sensitivity to light. As shown in Figure R1, the as-prepared N,S-C/In₂O₃ DHR also exhibited a certain degree of photocatalytic activities under green light (3W LED lamp, 525 nm) and yellow light (3W LED lamp, 590 nm). These results mean that the N,S-C/In₂O₃ DHR broadens the response range of light to wide wavelength. Figure R1 had been added in the revised supporting file as Figure S15, and corresponding descriptions have been added in the revised manuscript (page 13).

In addition, the N,S-C/In₂O₃ DHR could be easily recycled from the solution, due to the formation nature of assembling N,S-codoped carbon coated In₂O₃ ultrafine nanoparticles (~10 nm) into double-shelled hollow rods (~7 μm). The excellent photocatalytic performance of N,S-C/In₂O₃ DHR is benefited from its novel structure. (1) The hollow structure can improve light harvesting by simultaneously increasing the reflection on the outer shell and the diffraction on the hollow cavity. (2) The N,S-codoped carbon layers, uniformly distributed on the surface of In₂O₃ nanoparticles, can guarantee highly efficient electron transfer and improve the separation of photo-generated electron-hole pairs. (3) The double shells, composed by N,S-codoped carbon coated In₂O₃ ultrafine nanoparticles, can provide more reactive sites for the photocatalytic reactions. We then compared the photocatalytic activity of N,S-C/In₂O₃ DHR with N,S-C/In₂O₃ NP, N-C/In₂O₃ NP, In₂O₃ DHR, In₂O₃ HD and commercial In₂O₃ to reveal the influence of composition and structure on photocatalytic activity, showing that N,S-C/In₂O₃ DHR is the best photocatalyst among them in terms of catalytic performance. We also expanded the substrate scope (Figure R2) using N,S-C/In₂O₃ DHR as the photocatalyst, once again presenting its high catalytic performance. Corresponding experimental results had been added as Figure 4d in the revised manuscript.

Figure R1 Yields from oxidative hydroxylation of phenylboronic acid using N,S-C/In₂O₃ DHR under different LED irradiation.

Figure R2. Reaction yields of various phenols obtained from oxidative hydroxylation of different arylboronic acids with and without N,S-C/In₂O₃ DHR as a photocatalyst.

2. The authors mentioned that their catalysts could easily be recovered. However, this means the catalysts have very poor colloidal dispersity. Undoubtedly, this should not be called as an advantage! If the catalysts cannot be well dispersed, how good can their catalytic activity be?

Response: Thanks for the comments. The dispersity is indeed an important property of photocatalysts, which cannot be neglected. The N,S-C/In₂O₃ DHR could be easily recycled from the solution, but this does not mean its poor dispersity. As shown in Figure R4, only a small fraction of N,S-C/In₂O₃ DHR settled during the first 4 hours without stirring, indicating the deposition process of N,S-C/In₂O₃ DHR was slow. On the other hand, during the photoreactions, the solutions with N,S-C/In₂O₃ DHR were continuously stirred with dynamoelectric stirrers. As shown in Figure R10, the N,S-C/In₂O₃ DHR well dispersed in solutions under continuous stirring. Moreover, easy recycling of N,S-C/In₂O₃ DHR is due to its own gravity, which is different from the magnetic photocatalysts. Compared to the magnetic photocatalysts, the N,S-C/In₂O₃ DHR has greater application prospects, since it is not restricted by stirring conditions (mechanical stirring or magnetic stirring). In order to

avoid potential misunderstanding, Figures R4 and R10 have been added in the revised supporting file as Figures S17 and S18, respectively. The corresponding descriptions have also been added in the revised manuscript (page 15).

Figure R4. Sedimentation of (a) commercial In_2O_3 NP and (b) N,S-C/ In_2O_3 DHR for different time without agitation.

Figure R10. Dispersivity of (a) commercial In_2O_3 NP and (b) N,S-C/ In_2O_3 DHR for different time with stirring.

3. I was trying to find the challenge that the current work aimed to address. In the abstract, it was claimed “Excellent catalytic activity, high stability and easy recovery are three key elements for fabricating an efficient photocatalyst”. However, this claim is too vague for judging the importance and difficulty of the challenge. What is the exact challenge that is addressed in this work? Why is it important? Why is still not addressed in previous studies?

Response: Thanks for the comments and suggestions. The challenge of this work is to develop a simple method for fabricating efficient photocatalysts with excellent catalytic activity, high stability and easy recovery at the same time. Preparing ultrafine metal oxide nanoparticles

uniformly coated by a carbon layer as efficient photocatalysts is an effective solution in achieving the goal. The uniformly coated carbon layer can act as photo-generated-electron acceptor to improve the separation of photo-generated electrons and holes. Recent studies reported that heteroatom-doped carbon could further improve the conductivity of carbon materials, which is beneficial for improving the transfer of photo-generated electrons (*Adv. Mater.* **2015**, *27*, 6021-6028; *Nano Energy* **2016**, *19*, 373-381; *J. Mater. Chem. A* **2017**, *5*, 22964-22969). Nanoparticles with small sizes may provide more active sites for light absorption and reactions. As abovementioned, there are two difficulties in synthesizing such kinds of photocatalysts. (1) It is difficult to uniformly introduce heteroatoms into carbon layers, especially introducing double heteroatoms. (2) Recycling ultrafine photocatalyst nanoparticles from solution is also difficult. Since traditional techniques have unavoidable disadvantages limiting their practical applications, developing a simple method to fabricate efficient photocatalysts with excellent catalytic activity, high stability and easy recovery at the same time has been highly sought after. In this work, we successfully synthesized double-shelled hollow rods assembled by N,S-codoped carbon coated ultrafine In_2O_3 nanoparticles (N,S-C/ In_2O_3 DHR) by using MIL-68-In as the template and BIT as the modulator in one step. The N,S-C/ In_2O_3 DHR possesses excellent photocatalytic activity, improved sensitivity to light (3W blue LED as light source), high stability (catalytic efficiency has not been reduced and the morphology and composition of catalysts have not changed after multiple cycles), and easy recovery (achieved by simple centrifugation) at the same time. In order to more clearly express the importance of this work, corresponding descriptions have been added into the abstract of the revised manuscript.

4. When different amounts of competitive or modulator ligand were involved, the morphology can be controlled (*J. Am. Chem. Soc.* 2011, *133*, 15506–15513; *Angew. Chem.* 2009, *121*, 4833 –4837). Would the shape change with different amounts of BIT? If not, the effect of doping amount should be performed.

Response: Thanks for the useful suggestions. We agree with the reviewer's comments that the morphology of MOFs can be controlled when different amounts of competitive or modulator ligand were involved. According to the reviewer's suggestion, we changed the dosage of regulator (BIT) to control the morphology of N,S-codoped MIL-68-In. As shown in Figure R11, after changing the amount of BIT, the obtained N,S-codoped MIL-68-In could maintain the rod morphology. When further increasing the amount of BIT, the length-width ratio of N,S-codoped MIL-68-In rods gradually decreased. Figure R10 has been added in the revised supporting file as Figure S1. The corresponding descriptions (page 5) and references (*J. Am. Chem. Soc.* **2011**, *133*, 15506–15513 as ref. 36; *Angew. Chem. Int. Ed.* **2009**, *121*, 4833 –4837 as ref. 37) have been added in the revised manuscript.

Figure R11. N,S-codoped MIL-68-In rods with different widths prepared by using different amounts of BIT: (a-a₇) using 0.13 mmol (thin rods), (b-b₇) using 0.25 mmol (medium rods), and (c-c₇) using 0.36 mmol (thick rods).

5. Some information is lack of consistency. e.g. "Hence, annealing N,S-codoped MIL-68-In at 500 °C in a vacuum atmosphere with a ramping rate of 2 °C min⁻¹ was sufficient to ensure

complete conversion from MIL-68-In to In_2O_3 .” (line 109-111 in the main text) and “N,S-C/ In_2O_3 DHR was synthesized via the calcination of the obtained N,S-codoped MIL-68-In at 550 °C in vacuum atmosphere with heating rate of 5 °C min⁻¹ for 45 min.” (line 13-14 in the supporting information) What are the exact calcination temperature and heating rate?

Response: Thanks for the kind reminder. The exact calcination temperature, time and heating rate were 550 °C for 45 min with heating rate of 5 °C min⁻¹. We have corrected these information and unified corresponding descriptions in the revised manuscript (page 5 and supporting file page S1).

Reviewers' comments:

Reviewer #1 (Remarks to the Author):

The authors have addressed my concerns. The revised manuscript is acceptable for publication in my opinion.

Reviewer #2 (Remarks to the Author):

The authors well solved the issues in this revised edition, thus the acceptance for publication is recommended.

Reviewer #3 (Remarks to the Author):

I appreciate the authors' efforts in preparing the response letter which addresses most of the concerns from reviewers. Unfortunately, the manuscript was not well revised accordingly. I think the present version of manuscript is not publishable in Nature Communications but could be published in a more suitably revised form. Detailed comments are listed below:

1. The authors mentioned two bottlenecks of the area in the response letter: 1) It is difficult to control the coating of carbon layer and introduce the heteroatoms; 2) It is also difficult to recycle the ultrafine photocatalyst nanoparticles from solution. However, the first point was not mentioned in the manuscript at all!

2. "Easy recovery" has been claimed as one the most important advantages of the new catalysts in this study. Ultrafine photocatalyst nanoparticles are attractive because they have large surface areas and small diffusion distance for charge carriers to reach the surface active sites. It is straightforward to increase the size of catalysts for better recovery, which is exactly the strategy of the present study. However, this would lead to the drop of surface areas. The surface area of the catalyst is 37.6 m²/g, which is smaller than that of In₂O₃ nanoparticles and nanorods in previous studies (Ref. 13 & 15; Joule, 2018, 2, 1369-1381). How would the authors balance the ease of recovery with the surface area? Such information should be included in the introduction part of the manuscript.

2. Previous studies show that the construction of nanocrystal superstructures can improve the catalytic activity by enhancing the charge separation without sacrificing the surface area (Ref.15, Joule, 2018, 2, 1369-1381; Adv. Mater. Interfaces.2018, DOI: 10.1002/admi.201701694.). Nanocrystal superstructures should also improve the ease of recovery due to increased sizes as compared with single nanocrystals. What are the advantages of the current catalyst in comparison with In₂O₃ nanocrystal superstructures?

3. In the response to Q1 of my previous comments, the author claimed that a weaker light source (LED v.s. Xe lamp) was used in the present study, which suggested the catalysts exhibited higher sensitivity to light. This is not fair. A better comparison would be based on the quantum yield.

4. In the response to Q2 of my previous comments, the photos under stirring did not provide convincing evidences on the microscope dispersity of the catalyst. Dynamic light scattering results should be provided.

5. The response to Q3 should also be reflected in the introduction.

6. There are many grammar mistakes and wrong expressions, such as "What is need", "Special surface area"...

7. Besides oxygen vacancies, what are other defects? Such as hydroxide? If hydroxide is present, where is its peak in the O 1s XPS spectrum?

Response to Reviewer #1:

The authors have addressed my concerns. The revised manuscript is acceptable for publication in my opinion.

Response: We really appreciate the reviewer for the recommendation of publication.

Response to Reviewer #2:

The authors well solved the issues in this revised edition, thus the acceptance for publication is recommended.

Response: We really appreciate the reviewer for the recommendation of publication.

Response to Reviewer #3:

I appreciate the authors' efforts in preparing the response letter which addresses most of the concerns from reviewers. Unfortunately, the manuscript was not well revised accordingly. I think the present version of manuscript is not publishable in Nature Communications but could be published in a more suitably revised form. Detailed comments are listed below:

Response: We really appreciate the reviewer's useful comments and recommendation of publication in a more suitably revised form. We have carefully revised the manuscript based on the comments. We sincerely hope that the revised manuscript is now suitable for publication in the journal.

1. The authors mentioned two bottlenecks of the area in the response letter: 1) It is difficult to control the coating of carbon layer and introduce the heteroatoms; 2) It is also difficult to recycle the ultrafine photocatalyst nanoparticles from solution. However, the first point was not mentioned in the manuscript at all!

Response: Thanks for the kind reminder. According to the advice, the description of the first point has been added in the introduction of the revised manuscript (Page 3). The description is as follows: "It is difficult to control the coating of carbon layer and introduce the double heteroatoms into the carbon layer. In general, sufficient and intimate contact interface is the key factor to ensure the efficient transfer of photo-generated carriers. However, metal oxide nanoparticles are generally loaded on carbon materials, and only a small fraction of nanoparticle surface is in direct contact with the carbon materials. The uniform coating of carbon layers on metal oxide nanoparticles is still a challenging task. In addition, traditional techniques for introducing heteroatoms into carbon materials have unavoidable disadvantages, such as toxic precursors, special and/or sophisticated instruments, harsh conditions, and uneven distribution of doping atoms. Realizing the co-doping of double heteroatoms is even harder. Therefore, it is highly needed to overcome the challenge toward fabricating ultrafine metal oxide nanoparticles coated by co-doped carbon layers."

2. "Easy recovery" has been claimed as one the most important advantages of the new catalysts in this study. Ultrafine photocatalyst nanoparticles are attractive because they have large surface areas and small diffusion distance for charge carriers to reach the surface active sites. It is straightforward to increase the size of catalysts for better recovery, which is exactly the strategy of the present study. However, this would lead to the drop of surface areas. The surface area of the catalyst is $37.6 \text{ m}^2/\text{g}$, which is smaller than that of In_2O_3 nanoparticles and nanorods in previous studies (Ref. 13 & 15; Joule, 2018, 2, 1369-1381). How would the authors balance the ease of recovery with the surface area? Such information should be included in the introduction part of the manuscript.

Response: Thanks for the kind suggestion. As the reviewer commented, assembling the nanoparticles into photocatalysts with regular morphology and large size is straightforward to solve the problem of difficult recycling, but this solution would lead to the decrease of surface areas, reactive sites and photocatalytic activity. Thus, it is necessary to balance the ease of recovery with the surface area by improving other factors affecting photocatalytic activity. In this study, the N,S-codoped carbon coated In_2O_3 ultrafine nanoparticles were assembled into double-shelled hollow microrods. The as-prepared In_2O_3 photocatalysts exhibit excellent activity toward oxidative hydroxylation of a series of arylboronic acid substrates under blue-light irradiation. Benefiting from the improved light harvesting, resulted from simultaneously increasing the reflection on the outer shell and the diffraction on the hollow cavity of the double-shelled hollow structure, and the enhanced separation of photo-generated carriers due to uniformly coated N,S-codoped carbon layer, the recovery and surface area are well balanced. The corresponding description has been added in the introduction of the revised manuscript (Pages 3 and 4), and suggested paper has been added as Ref. 31.

3. Previous studies show that the construction of nanocrystal superstructures can improve the catalytic activity by enhancing the charge separation without sacrificing the surface area (Ref.15, Joule, 2018, 2, 1369-1381; Adv. Mater. Interfaces.2018, DOI: 10.1002/admi.201701694.). Nanocrystal superstructures should also improve the ease of recovery due to increased sizes as compared with single nanocrystals. What are the advantages of the current catalyst in comparison with In_2O_3 nanocrystal superstructures?

Response: Thanks a lot for inspiring comments. As compared to In_2O_3 nanocrystal superstructures, the current photocatalysts (N,S-C/ In_2O_3 DHR) have two additional advantages. (1) The N,S-C/ In_2O_3 DHR possesses double-shelled hollow structure, which can enhance light harvesting by simultaneously increasing the reflection on the outer shell and the diffraction on the hollow cavity. The enhanced light harvesting of N,S-C/ In_2O_3 DHR can generate more electrons and holes to participate in photocatalytic reactions. (2) The N,S-C/ In_2O_3 DHR is assembled by N,S-codoped carbon uniformly coated In_2O_3 ultrafine nanoparticles. The N,S-codoped carbon layer uniformly distributes on the In_2O_3 surface to

produce intimately contacted and maximized interfaces for the migration of photo-generated carriers. N,S-codoped sites could induce structure defects of the carbon framework and increase the electron delocalization, which further improve the separation of photo-generated carriers. Thus, the N,S-C/In₂O₃ DHR shows improved separation of photo-generated carriers and exhibits excellent photocatalytic activity. Moreover, the purpose of this study is to provide an effective approach to fabricate efficient photocatalysts assembled from codoped carbon uniformly coated metal oxide nanoparticles with excellent catalytic activity and easy recovery at the same time. The relevant discussions have been reflected in the abstract and introduction of the revised manuscript (Pages 3 and 4). The suggested papers have been added as Ref. 31 and 32.

4. In the response to Q1 of my previous comments, the author claimed that a weaker light source (LED v.s. Xe lamp) was used in the present study, which suggested the catalysts exhibited higher sensitivity to light. This is not fair. A better comparison would be based on the quantum yield.

Response: Thanks very much for the kind suggestion. We agree with the reviewer's opinion that the quantum yield is an important index for evaluating photocatalyst activity. Therefore, based on the literature report (*Appl. Catal. B: Environ.*, 2019, 242, 302), we have calculated the quantum yield of the catalysts by using the following formula:

$$N = \frac{E\lambda}{hc} = \frac{450 \times 10^{-9} \times 12 \times 3600 \times 2.35 \times 10^{-3} \times 4.27}{6.626 \times 10^{-34} \times 3 \times 10^8} = 9.80 \times 10^{20}$$

$$QY = \frac{\text{number of product}}{\text{number of incident photons}} = \frac{6.02 \times 10^{23} \times 0.1 \times 10^{-3}}{9.80 \times 10^{20}} = 0.0614$$

The catalyst solution was irradiated under blue LED ($\lambda = 450$ nm, 3W) for 12 h. The average light density was ca. 2.35 mW·cm⁻² and the irradiation area was 4.27 cm². Thus, the number of the incident photons (N) was 9.80×10^{20} . In the photocatalytic process, 0.1 mmol phenylboronic acid was used and completely converted into phenol after 12 h. Therefore, the quantum yield (QY) calculated was 0.0614.

By comparison with previous studies (Table R1), N,S-C/In₂O₃ DHR shows relatively high quantum yield, which might be attributed to its special structure (double-shelled hollow rods) and compositions (N,S co-doped carbon layer distributed on In₂O₃ nanoparticles). Related discussions have been added into the Supporting Information file (Page S13).

Table R1 Comparison for the photocatalytic quantum yield of different catalysts.

Catalysts	Light source	Photocatalytic reaction	Quantum yield	Reference
NiO/InTaO ₄	42.46 mW cm ⁻² Xenon lamp	CH ₃ OH	0.0120	Energy Environ. Sci. , 2011, 4, 1487
Au/CeO ₂	1.7 mW cm ⁻² LED	Oxidation of benzyl alcohol	0.0490	J. Am. Chem. Soc. , 2012, 134, 14526
g-C ₃ N ₄ /PDI/rGO _{0.05}	43.3 W m ⁻² Xe lamp (420 nm)	H ₂ O ₂ Production	0.0610	J. Am. Chem. Soc. , 2016, 138, 10019
g-C ₃ N ₄ /BDI ₅₀	Solar simulator (420 nm)	H ₂ O ₂ Production	0.0460	ACS Catal. , 2016, 6, 7021
NiO-In ₂ O ₃ /TiO ₂	200 W Mercury lamp	CO production	0.0104	Chem. Eng. J. , 2016, 285, 635
Co-ZIF-9/CdS	LED lamp (420 nm)	CO ₂ reduction	0.0193	Appl. Catal. B: Environ. , 2015, 162, 494
C ₃ N ₄	300 W Xe-lamp (420 nm)	H ₂ evolution	0.0500	J. Am. Chem. Soc. , 2017, 139, 3021
La/Cr-CaTiO ₃	500 W mercury lamp	H ₂ evolution	0.0241	Appl. Catal. B: Environ. , 2018, 225, 139
Zn ₃ In ₂ S ₆	300W Xe-lamp (420 nm)	H ₂ evolution	0.0402	Appl. Catal. B: Environ. , 2018, 232, 19
g-C ₃ N ₄	300 W Xenon lamp	H ₂ O ₂ Production	0.0430	Appl. Catal. B: Environ. , 2018, 232, 19
ZnO/BiVO ₄	300 W Xe lamp (450 nm)	O ₂ evolution	0.0500	Nano Energy , 2018, 51, 764
Mesoporous CoO	300 W Xe lamp (375 nm)	Reduction of Cr(VI)	0.0161	Appl. Catal. B: Environ. , 2018, 221, 635
ZnS/Ni	300 W Xe lamp (420 nm)	CO ₂ reduction	0.0560	Appl. Catal. B: Environ. , 2019, 244, 1013
SrBi ₄ Ti ₄ O ₁₅	300 W Xe lamp (365 nm)	CO ₂ reduction	0.0133	Nano Energy , 2019, 56, 840
N,S-C/In ₂ O ₃	3 W blue LED (450 nm)	Oxidative arylboronic acids	0.0614	This work

5. In the response to Q2 of my previous comments, the photos under stirring did not provide convincing evidences on the microscope dispersity of the catalyst. Dynamic light scattering results should be provided.

Response: Thanks a lot for the kind suggestion. In order to provide more convincing evidence, dynamic light scattering (DLS) has been used to characterize the dispersity of the catalysts. Figure R1 shows a plot with mean count rate as a function of different sedimentation time without (a) and with stirring (b). The count rate measured by DLS declines with the decrease of particle number in the solution [*Mol. Pharmaceutics* 2013, 10, 3392]. As shown in Figure R1a, when the samples were not under stirring, the count rate data of commercial In_2O_3 NP remained basically unchanged for 12 h. However, the data of N,S-C/ In_2O_3 DHR sharply decreased over 12 h. The results indicate that N,S-C/ In_2O_3 DHR can be easily recycled from the solution by its own gravity. When the samples were under continuous stirring for 12 h, the count rate data were almost unchanged (Figure R1b), indicating that the N,S-C/ In_2O_3 DHR can be well dispersed in solutions under stirring. These data correspond well with the photos of Figures S17 and S18.

In order to more clearly show the dispersity of the catalyst, Figure R1a and R1b have been added in Figures S17 and S18, respectively. Corresponding experimental details have been added to the supporting information file (Page S3).

Figure R1. Plots of mean count rate for different sedimentation time (detected by using dynamic light scattering). (a) Samples without agitation, and (b) samples under stirring.

6. The response to Q3 should also be reflected in the introduction.

Response: Thanks a lot for the kind suggestion. The response of Q3 has been reflected in the abstract and introduction of the revised manuscript (Pages 1, 3, and 4). Specific ideas are as follows: The challenge of this work is to develop a simple and effective approach to fabricate efficient photocatalysts assembled from codoped carbon uniformly coated metal oxide nanoparticles with excellent catalytic activity and easy recovery at the same time. However, there are two difficulties in synthesizing such kinds of photocatalysts. (1) It is difficult to uniformly introduce heteroatoms into carbon layers, especially introducing double heteroatoms. (2) It is difficult to balance the ease of recovery with the surface area. In this work, we have successfully synthesized double-shelled hollow rods assembled by N,S-codoped carbon coated ultrafine In_2O_3 nanoparticles (N,S-C/ In_2O_3 DHR) by using MIL-68-In as the template and BIT as the modulator in one step. The N,S-C/ In_2O_3 DHR possesses excellent photocatalytic activity and easy recovery at the same time.

7. There are many grammar mistakes and wrong expressions, such as "What is need", "Special surface area"...

Response: Thanks for the kind reminder. We have corrected these errors and proofread the entire manuscript to avoid such types and mistakes.

8. Besides oxygen vacancies, what are other defects? Such as hydroxide? If hydroxide is present, where is its peak in the O 1s XPS spectrum?

Response: Thanks very much for the useful question. Besides oxygen vacancies, the hydroxyl group assuredly exists in our samples. As shown in Figure 2c, The O 1s spectrum (Figure 2d) can be resolved into three peaks, assigning to lattice oxygen (O_L , 529.8 eV), oxygen-deficient region (O_V , 530.6 eV) and chemisorbed oxygen species (O_C , 531.8 eV). According to the reported references (*Appl. Phys. A: Mater. Sci. Process.*, 2008, 90, 317; *Joule*, 2018, 2, 1369-1381), the peak of O_C has been specifically interpreted as surface chemisorbed and dissociated oxygen species (e.g., hydroxyl species). Therefore, the peak of hydroxyl species should be the high binding energy (O_C). In order to more clearly express the O 1s spectrum, corresponding descriptions (Page 11) and references (*Joule*, 2018, 2, 1369-1381 as ref. 31, *Appl. Phys. A: Mater. Sci. Process.*, 2008, 90, 317 as ref. 45) have been added in the revised manuscript.

Figure 2c. High-resolution XPS spectra of O 1s.

REVIEWERS' COMMENTS:

Reviewer #3 (Remarks to the Author):

The authors made great improvement in this revised version which addressed all my concerns . I would like to suggest the publication of the current manuscript.